# A Framework for Human Corneal Endothelial Cell Culture and Preliminary Wound Model Experiments with a New Cell Tracking Approach

**DOI:** 10.3390/ijms24032982

**Published:** 2023-02-03

**Authors:** Francisco Bandeira, Gustavo Teixeira Grottone, Joyce Luciana Covre, Priscila Cardoso Cristovam, Renata Ruoco Loureiro, Francisco Irochima Pinheiro, Ricardo Pedro Casaroli-Marano, Waleska Donato, José Álvaro Pereira Gomes

**Affiliations:** 1Department of Ophthalmology, Federal University of São Paulo, São Paulo 04023-062, Brazil; 2Medicine School, Barcelona University, 08007 Barcelona, Spain; 3Biotechnology Post-Graduate Program, Potiguar University, Natal 59082-902, Brazil; 4Department of Surgery, Federal University of Rio Grande do Norte, Natal 59078-970, Brazil; 5Barcelona Tissue Bank, 08005 Barcelona, Spain

**Keywords:** corneal transplant, cell therapy, human corneal endothelial cells

## Abstract

Cell injection therapy is emerging as an alternative to treat corneal endothelial dysfunction (CED) and to avoid corneal scarring due to bullous keratopathy. However, establishing a standardized culture procedure that provides appropriate cell yield while retaining functional features remains a challenge. Here, we describe a detailed framework obtained from in vitro culture of human corneal endothelial cells (HCECs) and comparative in vivo experimental models for CED treatment with a new cell tracking approach. Two digestion methods were compared regarding HCEC morphology and adhesion. The effect of Y-27632 (ROCKi) supplementation on final cell yield was also assessed. Cell adhesion efficacy with two cell delivery systems (superparamagnetic embedding and cell suspension) was evaluated in an ex vivo human cornea model and in an in vivo rabbit CED model. The injection of supplemented culture medium or balanced salt solution (BSS) was used for the positive and negative controls, respectively. HCEC isolation with collagenase resulted in better morphology and adhesion of cultured HCEC when compared to EDTA. Y-27632 supplementation resulted in a 2.6-fold increase in final cell yield compared to the control. Ex vivo and in vivo adhesion with both cell delivery systems was confirmed by cell tracker fluorescence detection. Corneal edema and opacity improved in both animal groups treated with cultured HCEC. The corneas in the control groups remained opaque. Both HCEC delivery systems seemed comparable as treatments for CED and for the prevention of corneal scarring.

## 1. Introduction

Human corneal endothelial cells (HCECs) are responsible for keeping the cornea transparent by maintaining the tissue in a semi-dry state (~70% hydration) through the action of their ionic pumps and tight junction barrier [1]. Under normal conditions, HCECs have a negligible in vivo proliferation rate, likely due to the high aqueous humor concentrations of TGF-β2 and cAMP that upregulate p27KIP1, thus preventing HCECs from proliferating upon wounding [2,3]. Conversely, diseases such as corneal endothelial dystrophies and bullous keratopathy might compromise corneal endothelial function [4,5]. When the HCEC density falls below 500–700 cells/mm^2^, the cornea is at a high risk of decompensating, requiring endothelial corneal transplantation [6]; however, if the edema is chronic enough, it may eventually lead to stromal fibrosis secondary to bullous keratopathy, requiring penetrating corneal transplantation [7]. Currently, bullous keratopathy due to endothelial failure is a leading cause of corneal transplantation in many parts of the world, accounting for more than 40% of keratoplasty surgeries on some continents [8,9,10].

The current mainstay treatment for corneal endothelial failure is selective endothelial keratoplasty (EK) with the exchange of diseased cells for a healthy Descemet membrane–endothelial cell complex [5,11,12]. When compared to penetrating keratoplasty (PKP), EK techniques showed a significant improvement in patients’ quality of life, which can be attributed to a more straightforward postoperative regimen, faster recovery, and improved final visual acuity [5,13]. Even though EK has provided better refractive outcomes and an effortless postoperative follow-up, barriers to its adoption include increased cell loss and a steep learning curve [13]. Moreover, the demand for corneal transplants remains a challenging issue worldwide [14], which has become worse with the COVID-19 pandemic and increased life expectancy [15,16].

The research to unlock the growth potential of the corneal endothelium dates back to the 1980s, when Baum et al. first reported the mass production of cultivated HCECs from a 20-year-old donor [17]. Since then, numerous HCEC culture protocols have been reported with various combinations of isolation techniques ((collagenase, liberase, TrypLE, and ethylenediaminetetraacetic acid (EDTA)), extracellular matrices (chondroitin sulfate, laminin, fibronectin, and collagen coating), and growth factors (fibroblast growth factor, epidermal growth factor, nerve growth factor, and endothelial growth supplement) to achieve GMP-grade cells for clinical/therapeutic use [18].

Despite the extensive research, there is still no consensus concerning the most effective method for HCEC culture and delivery. In addition, several challenges that compromise HCEC quality and function (e.g., inter-donor variability [19,20,21] and endothelial–mesenchymal transformation (EnMT) [22,23,24]) have prevented most researchers from moving from preclinical research to human clinical trials [25].

Nevertheless, recent developments, including a peel and digest isolation approach, forced attachment, dual culture media, and Rho kinase inhibitor (ROCKi) supplementation, have successfully demonstrated the feasibility of achieving clinical-grade HCEC [24]. The introduction of ROCKi as a supplement to HCEC culture has also aroused researchers to investigate its in vivo regenerative properties, which have proven beneficial in surgeries that selectively remove the diseased endothelium (DSO/DWEK) [26,27]. The ultimate outcome of selective endothelial removal techniques is still controversial, with several cases resulting in very low endothelial cell count or failure [28]. Meanwhile, two cell delivery methods have shown satisfactory and safe outcomes in several studies involving animal models that have progressed to clinical trials, namely cell injection [29,30,31] and tissue-engineered endothelial keratoplasty [32,33].

Thus far, the cell injection therapy technique proposed by Kinoshita et al. relies solely on gravitational forces to promote cultured HCEC attachment to the recipient Descemet membrane (DM), which requires patients to withstand a long period in an uncomfortable prone position [34]. This method depends on patient compliance and is inherently subject to the inadvertent delivery of cells to areas that are completely outside of the targeted treatment (unwounded DM, iris, trabeculum, and lens) [35]; hence, a large number of cells (500,000/200 µL) need to be injected into the anterior chamber [36]. Iron microsphere or the nanoparticle embedment of CEC with magnetic delivery has been described as an alternative method to increase in vitro delivery efficiency 2.4-fold compared to gravity [37] without changing their canonical traits [38] while reducing corneal edema in animal models [39]. Nevertheless, confirming a superior CEC adhesion in vivo with magnetic delivery methods after surgery remains challenging and is yet to be proven.

The first clinical trial with cell injection supplemented with ROCKi in Japan unveiled satisfactory and fast patient recovery [34], with a survival rate of 90.9% after 5 years [40]. However, regulatory constraints in most countries impose the strict use of corneas that are unsuitable for clinical purposes or the peripheral remnants of from keratoplasty, which may not yield enough HCECs of good quality to restore corneal transparency [41]. HCEC culture research in Latin America is still embryonic. Within this context, this study aimed to establish a standardized HCEC culture system focusing on real-life conditions of a research lab in Latin America. We also investigated a wound model comparing HCEC attachment by magnetic attraction with the prone position (gravitational) and a novel cell tracking method.

## 2. Results

### 2.1. HCEC Isolation and Expansion

Isolation of HCECs was feasible and reproducible in both collagenase A and EDTA groups; there were no episodes of premature culture failure. The HCECs isolated with the collagenase yielded smaller cell aggregates which did not require pipetting and easily adhered to the uncoated acrylic well of the culture plates, forming cell clusters of different sizes (Figure 1A). Following isolation with EDTA, large aggregates of HCECs were present and underwent pipetting, after which they remained suspended as single cells or sphere aggregates and were unable to adhere to the uncoated culture plates. Upon exchanging the six-well culture uncoated plates for FNC-coated culture plates, HCECs isolated with EDTA could adhere and expand (Figure 1B). After 3 weeks in the culture system, the HCEC reached confluency in all samples, and a monolayered sheet of HCEC was identified. The HCECs were more homogeneously arranged and smaller, and they presented higher hexagonality in the collagenase group than the EDTA group (Figure 1C). In fact, HCECs treated with EDTA were markedly pleomorphic, with larger cells, and they did not present any hexagonality at all (Figure 1D).

### 2.2. Effect of Y27632 on Corneal Endothelial Cell Culture System

After 3 weeks under the culture system, all samples from both control and Y27632 groups yielded a confluent monolayered sheet of hexagonal HCEC. The final HCEC density obtained from samples submitted to treatment with Y-27632 ROCK inhibitor was significantly higher than in controls (Figure 1E)—1900 ± 349.5 and 715 ± 160.4 cells/mm^2^, respectively (*p* < 0.0001).

### 2.3. Cell Viability and Attachment after HCEC SPM Embedding

The TEM confirmed successful SPM embedding with dark vacuoles located within cultured HCEC cytoplasms (Figure 2A). We observed that SPMs were endocytosed into HCECs without affecting cell-wall integrity or cytoplasmatic volume, and there was no leakage of SPM to the extracellular space.

The mean absorbance level at 450 nm wavelength was marginally higher for the nanoparticle group (0.048 ± 0.01) than the control group (0.046 ± 0.00) (Figure 2B). This difference, however, was not statistically significant (*p* = 0.507) and supports the safety of SPM embedding in cultured HCECs.

After seven days, the negative control group did not show any fluorescence under the AF OCT confocal laser scan, thus confirming the absence of cultured HCECs. Conversely, for both NPT and positive control groups, the CLSO allowed the visualization of hyperreflective fluorescent points at the posterior layer of the wounded corneas (Figure 2C), which confirmed the successful attachment of the cultured HCECs. However, the pattern of cell attachment was indistinguishable among groups and did not respect any specific conformation. In addition, the cell clusters were sparsely distributed along the center and at the periphery, close to the limbus.

### 2.4. In Vivo Experimental Wound Models

Shortly after the endothelial wound creation, intense corneal edema appeared, and it was a common feature in SW and DMRx groups on the seventh postoperative day assessment. Nevertheless, the wound healing behavior differed completely between groups along with the postoperative follow-up. For instance, the corneal thickness change at the 30th postoperative evaluation was +41 ± 15 µm for the SW group (*p* < 0.005), whereas, for the DMRx group, this difference was more significant, resulting in a +1375 ± 180 µm difference (*p* < 0.0001). Moreover, the animals from the SW group appeared to have fully recovered endothelial function with completely transparent corneas and no signs of scarring at the final slit-lamp evaluation, 30 days postoperatively. The rabbits from the DMRx group, on the other hand, showed characteristic signs of posterior scarring and fibrosis. The results of the experimental wound models are summarized in Figure 3.

### 2.5. Cultured HCEC Injections in Rabbits

Corneal endothelium functional recovery after surgery was achieved exclusively in the groups submitted to injection of cultured HCECs (groups NANO30 and HCEC120). However, there were marked differences regarding intraocular inflammation and the resolution of corneal edema among them. Half of the animals in the NANO30 group presented with complete corneal transparency, while the other half presented with partial improvement of corneal clarity (2+/4) at the final evaluation. In the HCEC120 group, edema was only resolved partially in 75% of the animals (between 1+ and 3+/4), whereas one rabbit developed bullous keratopathy despite HCEC injection. Anterior chamber reaction occurred in both groups but presented more frequently in the NANO30 (with 2+/4 cells present in two animals and fibrin formation in one) compared to HCEC120, in which an intense fibrinotic reaction ensued in a single animal. There was no improvement in corneal edema in both the positive and the negative control groups, with rare inflammatory reaction; an intense fibrinotic reaction in the anterior chamber was present in one animal submitted to BSS injection. Representative images can be appreciated in Figure 4.

### 2.6. Fluorescence Tracking of Adherent Cultured HCEC after Injections

The in vivo tracking with Vybrant Green was notably effective, as the fluorescence of labeled cultured HCEC could easily be discriminated by the CSLO analysis. Distinctive fluorescence was detected centrally in the corneas of the NANO30 group (Figure 4D). In the HCEC120 group, the signal was heterogeneously distributed along the periphery and central area (Figure 4G), whereas it was completely absent in both control groups (Figure 4J,M).

## 3. Discussion

This study aimed to report the initial proceedings in setting up an endothelial cell culture lab based in South America. First, we sought to determine feasible means of isolating and seeding HCECs with two different protocols in such a way as to maintain their phenotypical properties and ensure optimal cell attachment. In sequence, we demonstrated that the final HCEC density yield achieved with supplemented Opti-MEM and Y27632 was superior to cells cultured with supplemented Opti-MEM only. Furthermore, we demonstrated with the XTT assay that cultured HCECs embedded with SPM nanoparticles did not affect their viability or morphological features. Next, to challenge the efficacy of cultured cell injection, we established an in vivo wound model in rabbits that successfully mimicked corneal endothelial dysfunction by either removing the DM or scraping off HCECs. Lastly, we compared the efficacy of injecting cultured HCECs with and without embedded SPM nanoparticles in the scrape wound model.

Although the success of HCEC expansion is a multifactorial issue, it is well known that precarious conditions such as extended corneal storage time and high donor age are often associated with culture failure [19,42]. We mitigated these problems by restricting the donor age to 50 and procuring these corneas within 5 days. Since the Sorocaba eye bank is less than 1 h away from the lab, we managed to set up a direct line to ensure fast and prompt retrieval of the corneas, which we deemed crucial for the success of HCEC expansion. The method of choice for isolating the DM containing HCECs influences the success just as well. There is an indirect correlation between donor age and DM adhesion to the stroma [43,44,45], which increases cell loss during its retrieval due to excessive manipulation and tears [46,47,48]. The donor age of ≤50 years as a cutoff did not pose any challenge regarding the retrieval of DM explants because all samples were prepared by a highly experienced surgeon (N.P.) with a validated atraumatic “no-touch” technique [49].

Several methods have been reported to isolate and expand HCECs. Initially, collagenase and trypsin were used in association with a selective l-valine-free medium to avoid keratocyte contamination [50]. However, this is a cumbersome process and time-consuming technique that over time has been replaced by more reproducible techniques such as “peel and digest” [20], sphere-forming assay [51], or serial explants [52]. Culture contamination with endotoxins or keratocytes is a serious concern while considering HCEC culture protocols. Both endotoxins derived from the ocular surface bacterial flora and keratocytes can delay cell proliferation and affect cell viability [53]. In addition to these adverse effects, because of the HCEC slow cycle behavior, keratocyte contamination can lead to a total failure of the culture system as they turn into highly proliferative fibroblasts that can overtake the culture completely [54]. All of the isolation techniques mentioned above are at increased risk of fibroblastic contamination [47]. However, we found that the “peel and digest” was more reproducible in our lab. Using this technique, we avoided the trabecular mesh transition and isolated a smooth DM, free of any stromal remnants and potential keratocytes. In order to counteract the possible deleterious effects of endotoxins, we left the explant for at least 12 h in 10% FBS. FBS has been shown to stabilize cell membranes and neutralize endotoxins by forming complexes or other lipid–lipid interactions [53].

The proliferation of HCECs is only possible after the release of contact inhibition blockage, which is popularly achieved by enzymatic digestion with dispase, EDTA/trypsin, or collagenase [20,47,55]. The latter two digestion methods are widely used in HCEC culture protocols [56], while the dispase method has been deemed more time-consuming and harsher for HCECs, resulting in frequent cell disintegration [55]. Our experiments showed that low-concentration collagenase A for a brief period was enough to produce viable cell aggregates (Figure 1) that seamlessly attached to culture dishes. At confluence, they presented as a monolayer of hexagonal cells, specific canonical traits of healthy HCECs [57]. Conversely, the isolation with EDTA/trypsin proved hard to reproduce because of the pipetting maneuvers that seemed to impair cell attachment. This method yielded cells that missed the canonical hexagonal shape of HCECs. Instead, their morphology resembled fibroblastic cell types (elongated and spindle-shaped), typical of EnMT [58]. Cell sorting with pipettes has been described as a common cause of mechanical stress [55] and a risk factor for EnMT [58]. Flame-polished pipetting required extensive manipulation of the cell aggregates, possibly damaging the HCEC membrane, which may explain the poor morphological quality.

In order to re-establish the functional capacity of the corneal endothelium, a minimum number of cultured HCECs need to be delivered onto the wounded area so as so to repopulate and recover the corneal transparency [30]. A minimum threshold of 700 cells has been reported as critical to maintaining corneal deturgescence [1]; however, when it comes to HCEC injection therapy, a much higher number of cells is desired. Okumura et al. recommended that a minimum of 500,000 cells should be injected into the anterior chamber to account for cell loss secondary to reflux, poor adhesion, or apoptosis [36]. The initial results of HCEC expansion demonstrated herein were far from meeting these minimal requirements (~800 cells/mm^2^). However, once we added the quintessential ROCK inhibitor Y27632, our cell yield surpassed the 2000 cells/mm^2^ threshold in 50% of our experiments, and none of the cultures resulted in a cell count below 1500 cells/mm^2^. In addition, a myriad of intra and extracellular events have been linked to the Rho downstream pathway. Of particular interest to HCEC culture, cell adhesion is regulated by the Rho/ROCK/myosin light-chain signaling cascade [36], while proliferation and viability have been linked to up- and down-regulation of p27KIP and cyclin D, respectively [59,60]. Extensive research on Y27632′s effects on HCEC culture has shown invaluable benefits such as an increase in cell attachment rates, improvement of cell viability, and prevention of EnMT [36,59,60], all in line with our findings.

Before initiating the functional experiments with rabbits, we tested whether superparamagnetic particles could be embedded to our cultured HCEC without affecting viability. We chose 100 nm superparamagnetic beads (Fluid-Mag Lipid) as a function of its near-irrelevant toxicity profile previously shown in ARPE-19 [43] and human endothelial cells [37,38,61]. These findings were also valid for the cultured HCECs, as shown with the very low absorbance on our XTT experiment. We also sought to address cell tracking using a novel noninvasive method relying on Vybrant Green staining. This accessible and safe cell-permeable DNA dye emits a 488 nm fluorescence that can easily be detected by the CLSO available in several imaging devices, commonly used in routine ophthalmology practice [62]. We proved the applicability of this method in both our ex vivo attachment and our in vivo therapeutic experimental models. Xia et al. also described a similar method of cultured HCEC fluorescence tracking [63]. However, they used a viral vector with green fluorescent protein lentivirus, which may theoretically lead to DNA integration to the recipient and other diseases [64].

Lastly, after electing the scrape wound as the appropriate model for challenging the functional aspect of our HCEC culture protocol, we compared edema resolution with cultured HCECs supplemented with Y27632 delivered by gravity or magnetic attraction (SPM-embedded HCEC). We deliberately decided to employ a very short follow-up time (only 4 days), due to our (unpublished) and other previous observations from experiments in rabbit models with endothelial injury, in which the maximal edema was found at this timepoint [31,36,63,65]. In this way, we avoided the influence of the natural regeneration capacity of the rabbit endothelium, which, at this early stage after trauma, seems not to have started yet. A noteworthy comment regarding cell tracking is that the attached cells in the nanoparticle were located centrally within the targeted wound area. The HCEC group without magnetic nanoparticles showed a more heterogeneous distribution, which is a concern if translated to clinical application because it can affect the treatment efficacy or recovery speed. At the final examination, we concluded that injection with both HCECs alone and SPM-embedded HCECs was able to clear the corneas either partially or totally, while corneal edema remained in both control groups. Nevertheless, the presence of fluorescence emitted by Vybrant Green even after 4 days confirms that cultured cells from both groups were able to attach to the DM successfully. In addition to preventing corneal scarring, the improvement of corneal clarity and edema in both treatment groups is a sign that the injected cells remained viable, but they were also satisfactorily functional.

Over half of the corneas procured for transplantation (55%) came from the northern hemisphere (India or the United States). Conversely, most countries in Latin America are not self-sufficient when it comes to providing corneal tissue for transplantation [14]. The recent devastating 2019 COVID-19 pandemic crisis aggravated the already alarming problems with eye tissue donation [16]. In particular, the uncertainty regarding the risk of virus transmission via cornea motivated eye banks to temporarily cease activity for a considerably long period (weeks to months) [15,16]. It is likely that the global shortage of donors will deteriorate severely in the aftermath of the COVID-19 pandemic. Considering the current situation, we adapted a lab to investigate the feasibility of starting HCEC culture as an alternative for keratoplasty. We acknowledge several shortcomings with our current report, including a lack of specific HCEC genetic and phenotypical biomarker characterization, absence of functional gene and protein expression related to endothelial pump function, pooling of donor corneas, a short follow-up, and limited sample size of the experimental models.

Nonetheless, even with the scarce infrastructure and access to resources, we could still start a successful HCEC expansion protocol, which proved efficient after HCEC injection in an animal model. In addition, we presented a simple and affordable method for cell tracking relying on a device that is already approved for clinical use (OCT). To our knowledge, this approach is the first of its kind and may be further improved to facilitate objective comparisons of cell adhesion efficacy between different delivery methods for CED (eg. counting of adherent cells). Of course, additional work is obligatory before advancing to cell therapy in humans. Compliance of the cell yield, quality, and functional properties of HCEC cultured with this same protocol at a certified GMP eye bank will be the subject of a follow-up study (manuscript in preparation).

## 4. Materials and Methods

This study was conducted at the Ocular Surface Advanced Center of the São Paulo Federal University, Brazil. The Sorocaba Eye Bank (BOS) and São Paulo Hospital Eye Bank (BOHSP) provided research-grade corneas [*n* = 20] with written consent from the donor’s next of kin authorizing usage for research purposes. All corneas included in the study did not meet the criteria for clinical application and presented an endothelial cell count higher than 2000 cells/mm^2^. Furthermore, postmortem preservation time was below 6 h (fresh cadavers) or below 24 h (under refrigeration); the time under preservation was below 10 days, and the donor age below 50 years old.

Corneas from donors diagnosed with sepsis, consumptive diseases, or severe trauma were automatically excluded. Excluding factors upon corneal biomicroscopic evaluation included signs of prior ocular surgery or infection. All tissues were preserved in Optisol-GS (Alchimia, Firenze, Italy) at 4 °C before use.

### 4.1. Isolation and Culture of Human Corneal Endothelial Cells

Twenty corneas were randomly selected for the isolation and expansion experiments. First, the preservation flask was left at room temperature for at least 2 h before manipulation. The corneas were then rinsed with DMEM containing 50 mg/mL gentamicin and 1.25 mg/mL amphotericin B. Next, the DM was stripped, and each explant was carefully placed in a 15 mL Falcon tube filled with 12 mL of Opti-MEM, stored overnight at 37 °C and 5% CO_2_. The DM containing HCEC was then removed from the incubator and hemi-sectioned with a corneal scissor. The half of each cornea was placed in either a single well of a six-well plate containing culture medium supplemented with either collagenase A (1 mg/mL) for a period of 12 h or a 15 mL Falcon tube containing 6 mL of 0.03% EDTA for 50 min. After digestion, the HCEC aggregates were collected by centrifugation at 2000 rpm for 3 min to exchange the digestion solution for Opti-MEM I culture medium (InVitrogen-Gibco, São Paulo, Brasil), supplemented with 5 ng/mL epidermal growth factor (EGF) (Biomedical Technologies Inc., Stroughton, MA, USA), 0.5% dimethyl sulfoxide, 5 μg/mL insulin, 5 μg/mL transferrin, 5 ng/mL selenium, 0.5 μg/mL hydrocortisone, 1 nM cholera toxin, 50 μg/mL gentamicin, and 1.25 μg/mL amphotericin B [55].

Following centrifugation, larger HCEC aggregates were pipetted through a flame-polished glass Pasteur pipette into single cells while small clusters were left to attach spontaneously. Finally, cells from each sample were seeded at a single well of either an FNC-coated or an acrylic six-well plate filled with culture medium and stored at 37 °C and 5% CO_2_. Medium changes were carried out twice a week, and the samples were checked until confluency over 80% was achieved.

The final endothelial cell density was measured under 200× magnification with an optical microscope (Zeiss Axioplan, Göttingen, Germany). A qualitative cell analysis was also performed regarding their cell size and hexagonality.

### 4.2. ROCK Inhibitor Treatment

The expansion of HCECs under ROCKi (Y27632; StemCell Technologies INC, Vancouver, BC, Canada) was carried out after standardization of the HCEC isolation. Collagenase A (1 mg/mL) digestion approach was the least toxic and most effective digestion method; hence, it was chosen for all subsequent experiments. Ten corneas (*n* = 10) were randomly selected for this experiment. The samples were hemi-sectioned with a corneal scissor and divided into two groups, A and B. For all the samples in group A, the isolation before HCEC culture followed the steps exactly as described above, while samples from group B were submitted to supplementation with 10 µM Y27632 during the seeding step. At the end of 3 weeks, endothelial cell density was assessed as stated above, and the results were compared.

### 4.3. HCEC Superparamagnetic Embedding

On the basis of our group’s previous experience with ARPE cell delivery [66], cultured HCECs were submitted to treatment with a superparamagnetic solution (SPM), FluidMag Lipid^®^ (Chemicell, GmBH, Berlin, Germany). This solution uses 100 nm diameter lipid-coated ferumoxide nanoparticles, which can manipulate cell delivery through magnetic-field guidance. In brief, HCECs were cultured for 2 weeks until they reached 50% confluency. At this point, the cells were dissociated as per the standardized isolation method and subsequently seeded at a cell density of 5 × 10^3^ in 12 wells (*n* = 12) of an FNC-coated 96-well plate. Cultured HCECs were then divided into two groups: control (*n* = 6), which did not receive any further treatment, and the nanoparticle HCEC group (*n* = 6), which was incubated with SPM. First, the samples were incubated with 0.1 mg/mL Fe-diluted SPM for 4 h; then, the cultured medium was removed, and each well was rinsed twice with PBS, followed by centrifugation as described in Section 2.1.

Endothelial cell viability was tested using the XTT assay (TOX-2 KIT, Sigma-Aldrich, Darmstadt, Germany) as previously described [32]. Then, a magnetic guidance test was performed. HCECs were retrieved with collagenase A (1 mg/mL) and exposed to an Nd magnetic rod of 3500 Gauss (0.35 Tesla). A positive result was assigned if cell cluster displacement occurred.

The successful embedding of HCEC with the SPM solution was evaluated by transmission electron microscopy (TEM) to confirm intracellular iron content. In brief, for this analysis, cultured HCECs were seeded in glass slides; after 24 h of SPM embedding they were fixed with 4% paraformaldehyde/1% glutaraldehyde in 0.1 M phosphate buffer, stained with 0.5% uranyl acetate (en bloc), and then dehydrated in graded alcohol concentrations. Samples were transferred to propylene oxide and were embedded in Araldite resin. Ultrathin sections were collected on uncoated G300 copper grids and stained with 1% aqueous phosphotungstic acid and uranyl acetate. Sections were examined with a transmission electron microscope operating between 80 and 120 kV (JEOL 1200 EX II; JEOL, Tokyo, Japan) equipped with a digital camera (GATAN 832, Pleasanton, CA, USA).

### 4.4. Fluorescence Confirmation of Cultured HCEC Attachment to Recipient Corneas: Ex Vivo Wound Model

Cultured HCECs were labeled with Vybrant^®^ DyeCycle^®^ Green cell tracker (Thermo-Fisher, Waltham, MA, USA) before SPM embedding. Briefly, the medium from HCEC culture plates was exchanged for 3 µL of Vybrant Green diluted in 1 mL of fresh culture medium. The plates were incubated for 30 min at 37 °C and 5% CO_2_, and then covered in aluminum foil to avoid light exposure.

The samples were divided according to the type of cultured HCEC transplant treatment: SPM embedded cultured HCECs/nanoparticle transplant (NPT) group (*n* = 3); cultured HCEC not submitted to nanoparticle embedment (positive control) group (*n* = 3); no cultured HCEC transplant (negative control) group (*n* = 3).

For recipient corneas of all groups, the endothelial cells were removed using a previously described scraping method [67] with minor modifications. In brief, a sterile poly-vinyl alcohol sponge was used to gently scrape off the HCEC while maintaining the DM as intact and smooth. Next, samples were left to rest inside a positive laminar flow hood for approximately 6 h. Lastly, complete removal of native endothelial cells was confirmed with 0.05% Trypan blue staining.

The Vybrant Green-labeled HCECs were dispensed into a Proteosave container (Sumitomo Bakelite) to block nonspecific protein absorption (1 × 10^6^ cultured HCECs were distributed per 450 μL of modified SHEM). Cell delivery was performed under sterile conditions, and the recipient corneas remained mounted for at least 12 h, but a different method was applied for each group.

For the NPT Group, the corneas were mounted in a Barron artificial anterior chamber (Corza-Katena, Parsippany, NJ, USA) with the endothelial side facing downward and filled with modified SHEM. A sterile 3500 Gauss magnet rod was placed over the epithelium at the recipient cornea apex. Cultured HCEC injection was gradually performed through one of the anterior chamber ports. Conversely, the corneas from the positive control group were mounted at a donor punch with the endothelial side facing upward, while the cultured negative SPM HCECs were carefully inoculated onto the DM. The negative control group’s corneas were inoculated with modified SHEM that did not contain any cultured HCECs.

After 12 h, each cornea was washed with PBS, placed into a single well of a six-well plate, and incubated with the modified SHEM culture medium at 37 °C and 5% CO_2_. The corneas remained under culture for approximately 7 days, and the medium was replenished every 2 days before fixation with 4% PFA for 15 min at 4 °C. Following fixation, the samples were mounted at donor punch base with the endothelial side facing upward for evaluation with optical coherence tomography (OCT) (Spectralis, Heidelberg GmBH, Heidelberg, Germany). The confocal scanning laser ophthalmoscopy (CSLO) used in the autofluoresce detection mode of the OCT was set at 488 nm and 521 nm wavelengths to detect Vybrant Green for excitation and barrier phases, respectively.

### 4.5. In Vivo Endothelial Regeneration with Injected HCEC in White New Zealand Rabbits

All animals were handled according to the guidelines of the ethics committee of the Federal University of Sao Paulo (registration code 223161013), and the Association for Research in Vision and Ophthalmology Statement for the Use of Animals in Ophthalmic and Vision Research. A total of 22 male New Zealand White rabbits, 6 to 8 months old, weighing between 1.8 and 2.2 kg, were used for two different experiments, performed sequentially. In the first experiment, six rabbits were randomly assigned to two endothelial wound model groups; in the second experiment, another 16 animals were assigned to four treatment groups. Anesthesia was performed with 50 mg/kg ketamine and 10 mg/kg xylazine intramuscular injection.

#### 4.5.1. Endothelial Wound Model

In one group (*n* = 3), the rabbits were submitted to complete removal of endothelial cells by scraping a wound with the assistance of a soft-tip vitreoretinal cannula (DORC, Netherlands); this group was named “SW”. In short, two extra side-port wounds were created, through which 23 G trocars were placed at 3 and 9 o’clock to ensure stable manipulation of the endothelium with the soft-tip cannula. The endothelial cells were meticulously scraped off with care not to damage the DM. A solution of 0.05% Trypan blue (Ophthalmos, São Paulo, Brazil) was used to confirm the complete removal of all cells. Wherever the staining was incomplete, the scraping maneuvers were repeated at that site until the whole posterior cornea stained successfully. Three rabbits were submitted to the “descemetorhexis” technique, as originally described by Melles et al. [68]. This group was referred to as the “DMRx” group. Briefly, the anterior surface of the rabbits’ cornea was marked with a 7 mm trephine, and one extra 1 mm side-port wound was created, through which an inverted Sinskey hook (ASICO, Westmont, IL, USA) was used to detach the edges of the DM along the trephine marking. After the borders were released, the remainder of the DM was gently centripetally stripped off the posterior stroma, then pulled toward the side-port incision, and then removed from the anterior chamber, very carefully, to avoid trauma to the corneal stroma.

For both groups, the corneal incisions were closed with 10-0 nylon sutures (Ethicon, Johnson & Johnson, São Paulo, Brazil) followed by a subconjunctival injection of 0.3 mL of triamcinolone acetonide (4 mg/0.1 mL, Ophtaac^®^, Ophthalmos, São Paulo, Brazil); ciprofloxacin/dexamethasone drops (EMS, LeGrand, Brazil) were started immediately after the procedure. At the 7th and 30th days postoperatively, a slit-lamp examination was performed to analyze corneal opacity and fibrosis qualitatively, whereas the corneal thickness and posterior wound regularity were measured using an anterior-segment OCT (Visante^®^, Carl Zeiss, GmBH, Jena, Germany).

#### 4.5.2. Experimental Treatment with Cultured Cell Injections and Fluorescence Confirmation of HCEC Adhesion

Following the initial endothelial wound model experiments, the SW model was selected to perform the experimental treatment. The animals were separated into four groups according to the type of treatment: injection of cultured HCEC with nanoembedded paramagnetic particles + resting prone position for 30 min (NANO30, *n* = 4); injection of cultured HCEC + resting prone position for 120 min (HCEC120, *n* = 4); injection of balanced salt solution (negative control, *n* = 4); injection of culture medium (positive control, *n* = 4). The surgery was performed equally for all groups. After the anesthesia was achieved, a small lid speculum was placed on the left eye. Povidone/iodine was flushed over the ocular surface before incisions; three 0.5–1 mm side port incisions were created (superiorly, nasally, and temporally). The superior incision was used to place an anterior chamber maintainer; the other incisions were used for surgical maneuvers. The endothelial cells were scraped off the DM thoroughly with a soft-tip cannula; 1% Trypan blue was then used to confirm complete removal of the endothelium. In sequence, all side-port incisions were closed with an “X-stitch” mini-running suture, and 0.1 mL of culture medium and pooled cultured HCECs were slowly injected directly into the anterior chamber using a 27 G needle connected to a 1 mL syringe. After the full content in the syringe was inside the eye, a Weckcel sponge was used to cover the needle entrance to avoid reflux through the side-port. An adapted bandage contact lens was glued with 100% ethyl-2-cyanoacrylate (Loctite^®^, Henkel, Jundiaí, Brasil). A fixed combination of 0.3% gatifloxacin and 1% prednisolone acetate was applied topically immediately after the procedure and then q.i.d postoperatively until euthanasia was performed, 1 month after the intracameral injection. The contralateral eye served as a control. Clinical edema and inflammation were assessed daily for only 4 days using a portable slit lamp. In vivo cultured HCEC adhesion was assessed with CSLO after the euthanasia, followed by immediate eye globe enucleation from each group.

The isolation and adhesion of in vitro cultured HCECs, ex vivo attachment of SPM-embedded HCECs, and non-SPM cultured HCECs, as well as in vivo experiments, are summarized in Figure 5, Figure 6 and Figure 7.

### 4.6. Statistical Analysis

All numeric data obtained were expressed as the mean ± standard deviation (M ± SD). All statistical analyses were performed using Prism 6.0 (GraphPad Software, La Jolla, CA, USA). The comparison of HCEC final densities between the different isolation groups and the statistical comparisons between nontreated HCECs (hemi-sectioned control) and Y27632-treated HCECs were evaluated using paired-sample *t*-tests. All results with a *p*-value < 0.05 were deemed to be statistically significant.

## 5. Conclusions

In summary, establishing culture cell systems is a long, complex, and demanding task that involves standardization and safety precautions to ensure cell quality and to avoid complications. Nevertheless, we hope that this study will provide an initial framework that may lay the foundation for further research, even in small labs not specialized in endothelial cell culture. Furthermore, the discovery of a new cell tracking method may be of use to analyze and improve the efficacy of cell delivery systems for CED. Future work, including translational work using an eye bank at the center of cell culture proceedings, may also improve the success of the results shown here.

## Figures and Tables

**Figure 1 ijms-24-02982-f001:**
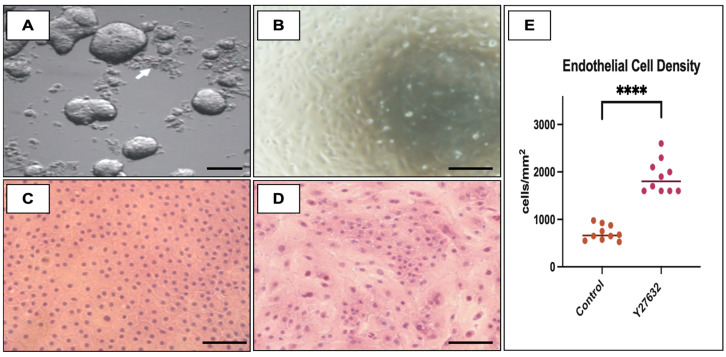
Human cornea endothelium isolation and attachment. Cell attachment after isolation with modified SHEM culture media (mSHEM) + collagenase A (1 mg/mL) for 12 h (**A**, 200×), and isolation with mSHEM + 0.03% EDTA for 50 min (**B**, 100×). Collagenase isolation resulted in small HCEC aggregates that successfully adhered to the uncoated dishes (**A**, white arrow), while EDTA isolation resulted in large aggregates that required pipetting to obtain single cells that only adhered to coated plates (**B**). Endothelial monolayer of HCEC morphology after 80% confluence (21 days) with HCEC isolated with collagenase A (**C**, 200×), and EDTA (**D**, 200×). HCECs cultured after collagenase isolation resulted in a homogenous, hexagonal, and compact cell mosaic resembling the typical human endothelium (**C**), while the monolayered endothelium resulting from EDTA isolation was disorganized, heterogeneous, and pleomorphic with rare spindle-shaped cells resembling fibroblasts (**D**). (**E**) Endothelial cell density after culture with and without ROCK inhibitor (**** *p* < 0.001).

**Figure 2 ijms-24-02982-f002:**
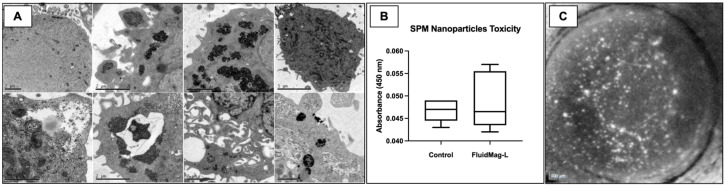
Cultured HCEC viability and adhesion to human DM after superparamagnetic particle embedding. (**A**) TEM images confirming SPMs endocytosis with visible particles inside HCEC cytoplasm and at cell wall surface, cell-wall integrity is preserved with no leakage of SPM or organelles to the extracellular space. (**B**) Cell absorbance level at 450 nm wavelength for HCECs cultured without superparamagnetic particles compared to FluidMag-embedded cultured HCECs. (**C**) OCT infrared image demonstrating green, fluorescent HCEC adhered to DM 7 days after ex vivo attachment assisted with a 3500 Gauss magnet. Scale bars: (**A**). Top row, from left to right: 2 µm, 2 µm, 2 µm, and 5 µm. Bottom row, from left to right: 1 µm, 2 µm, 2 µm, and 1 µm.; (**C**). 200 µm.

**Figure 3 ijms-24-02982-f003:**
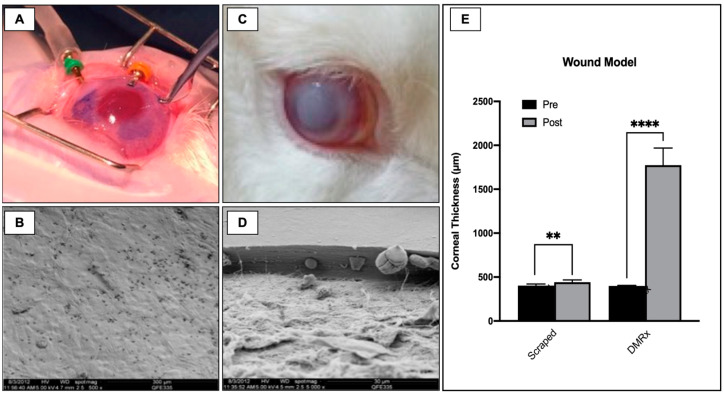
In vivo experimental wound models. (**A**) Trypan blue staining after scraping maneuvers confirming native CEC removal (External photograph with digital smartphone camera, 1.4× zoom). (**B**) Scanning electron microscopy (500×) showing a DM devoid of endothelial cells. (**C**) Sustained corneal opacity 30 days after descemetorhexis in rabbit corneas (6×). (**D**) Scanning electron microscopy (5000×) showing the bare posterior stroma after DM removal. (**E**) Bar chart comparing the corneal thickness after scraping and descemetorhexis at the 30th postoperative day. ** *p* < 0.01; **** *p* < 0.001.

**Figure 4 ijms-24-02982-f004:**
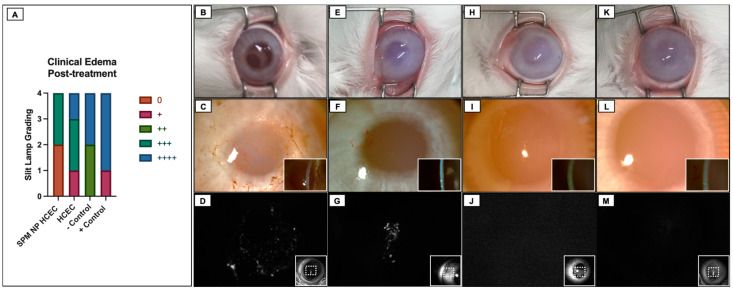
In vivo cultured HCEC injection. (**A**) Cumulative graph of clinical opacity grading according to Sotozono’s grading. Upper row, macroscopic image of rabbits’ cornea at the 4th day postop (6×). Mid row, diffuse slit-lamp image (16×), with inserts showing a narrow-slit beam (10×) at the 4th day postop. Lower row, CLSO after enucleation (7th day postop), with inserts showing the infrared macroscopic image of the cornea, where the dotted lines represent the area of interest shown in the larger pictures. (**B**–**D**) Representative image of rabbits submitted to SPM embedded cultured HCECs (*n* = 4). (**E**–**G**) Representative image of rabbits submitted to cultured HCECs without magnetic particles (*n* = 4). Fluorescent HCECs are seen as bright spots in figures centrally in (**D**), and diffusely scattered in (**G**). (**H**–**J**) Representative image of rabbits submitted to BSS injection (negative control, *n* = 4). (**K**–**M**) Representative image of rabbits submitted to culture media supplemented with Y-27632 (positive control, *n* = 4).

**Figure 5 ijms-24-02982-f005:**
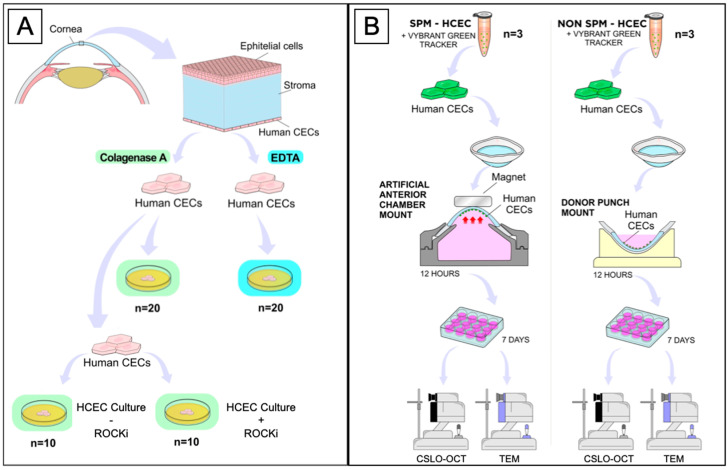
Illustrative summary of in vitro and ex vivo experiments. (**A**) In vitro isolation, adhesion, and culture of HCECs. (**B**) Adhesion of FluidMag-embedded cultured HCECs and nonmagnetic cultured HCECs to ex vivo human corneas. Legend: CECs, corneal endothelial cells; EDTA, ethylenediaminetetraacetic acid tetrasodium salt; ROCKi, Rho kinase inhibitor (Y27632); SPM, superparamagnetic particles; CLSO-OCT, confocal laser scanning ophthalmoscopy; OCT, optical coherence tomography; TEM, transmission electron microscopy.

**Figure 6 ijms-24-02982-f006:**
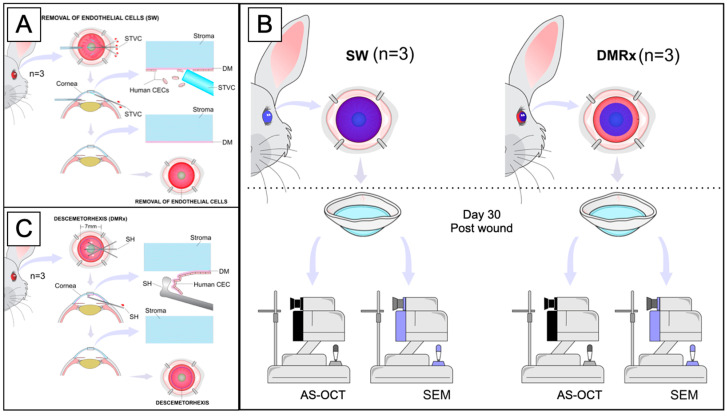
Illustrative summary of in vivo wound model. (**A**) Depiction of descemethorexis wound model. (**B**) Depiction of scraping wound model. (**C**) Confirmation of HCEC removal with Trypan blue staining and microscopy analysis. Legend: STVC, soft-tip vitreoretinal cannula; DM, Descemet membrane; CECs, corneal endothelial cells; SH, Sinskey hook; AS-OCT, anterior segment optical coherence tomography; SEM, scanning electron microscopy.

**Figure 7 ijms-24-02982-f007:**
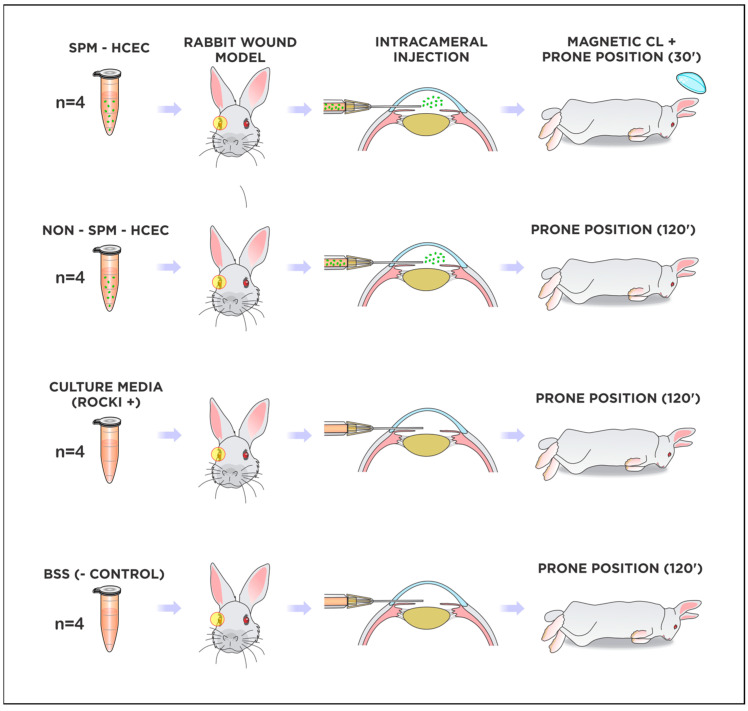
Illustrative summary of in vivo experimental model of cultured HCEC injection into rabbits. Legend: HCEC, human corneal endothelial cells; SPM, superparamagnetic particles; CL, contact lens; ROCKi, Rho kinase inhibitor (Y27632); BSS, balanced salt solution.

## Data Availability

Not applicable.

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
