# Peer review of "A Framework for Human Corneal Endothelial Cell Culture and Preliminary Wound Model Experiments with a New Cell Tracking Approach"

_ijms, 2023, doi:10.3390/ijms24032982_

Round 1

Reviewer 1 Report

The human corneal endothelial cell culture and preliminary wound model investigations were carried out by the authors. As the writers indicated, there was nothing unique in the manuscript other than it being the first of its sort from Latin America.

In addition, the manuscript is poorly written.

Author Response

Reply: We have submitted the manuscript to a professional native English speaker certified review. The changes were mostly restricted to punctuation rules and other conventions. No major changes were made regarding flow or clarity by the certified MDPI english Reviewer.    The link for the certificate follows below. https://www.dropbox.com/s/hbya3f0iijo3iro/English-Editing-Certificate-57460.pdf?dl=0

Reviewer 2 Report

Initial Framework for Human Corneal Endothelial Cell Culture and Preliminary Wound Model Experiments.

Line 32. It reads: “The human corneal endothelial cells (HCECs) are responsible for keeping the cornea transparent by maintaining the tissue at a semi-dry state (~70% hydration) through the action of their ionic pumps and tight junction barrier[1].. HCEC have negligible in vivo proliferation rate[2]., likely due to the aqueous humor high concentrations of TGF-2 and cAMP that up-regulate p27KIP1, thus preventing HCEC from proliferating then regenerate upon wounding[3, 4].”

COMMENT

It is true than under normal circumstances HCEC have negligible in vivo proliferation rate, but in special conditions it has been shown that the replication can increase.

Three of the cited references (2-4) were published 19 or more years ago, therefore, for this reason, they can actually be considered outdated.

Additional suggested references:

Galvis V, Tello A, Gutierrez ÁJ. Human corneal endothelium regeneration: effect of ROCK inhibitor. Invest Ophthalmol Vis Sci. 2013 Jul 24;54(7):4971-3. doi: 10.1167/iovs.13-12388. PMID: 23883789.

Zavala J, López Jaime GR, Rodríguez Barrientos CA, Valdez-Garcia J. Corneal endothelium: developmental strategies for regeneration. Eye (Lond). 2013 May;27(5):579-88. doi: 10.1038/eye.2013.15. Epub 2013 Mar 8. PMID: 23470788; PMCID: PMC3650267.

Klyce SD. Endothelial pump and barrier function. Exp Eye Res. 2020 Sep;198:108068. doi: 10.1016/j.exer.2020.108068. Epub 2020 Jul 11. PMID: 32663497.

Van den Bogerd B, Dhubhghaill SN, Koppen C, Tassignon MJ, Zakaria N. A review of the evidence for in vivo corneal endothelial regeneration. Surv Ophthalmol. 2018 Mar-Apr;63(2):149-165. doi: 10.1016/j.survophthal.2017.07.004. Epub 2017 Aug 4. PMID: 28782549.

Line 38. It reads: “When HCEC density falls below 500-700 cells/mm, the cornea is at a high risk of decompensating and eventually leads to blindness due to corneal fibrosis secondary to bullous keratopathy[1, 7].”

COMMENT

It would be very useful for the reader to point out here that in various areas of the world, endothelial failure is among the first indications for corneal transplantation.

Consider modifying to: “When HCEC density falls below 500-700 cells/mm, the cornea is at a high risk of decompensating requiring an endothelial corneal transplantation, but if the edema is chronic enough eventually may lead to stromal fibrosis secondary to bullous keratopathy, and will require a penetrating corneal transplantation. [1, 7]. Currently bullous keratopathy due to endothelial failure is one of the first indications for corneal transplantation in many parts of the world.”

References to be cited:

Galvis V, Tello A, Laiton AN, Salcedo SLL. Indications and techniques of corneal transplantation in a referral center in Colombia, South America (2012-2016). Int Ophthalmol. 2019 Aug;39(8):1723-1733. doi: 10.1007/s10792-018-0994-z. Epub 2018 Jul 25. PMID: 30047076.

Zhao S, Wan X, Yao W, Xu J, Le Q. Trends of Corneal Transplantation in Adults from 2010 to 2019 in East China: A 10-Year Experience. Ophthalmic Res. 2022;65(1):30-39. doi: 10.1159/000519824. Epub 2021 Sep 29. PMID: 34587612.

Chilibeck CM, Brookes NH, Gokul A, Kim BZ, Twohill HC, Moffatt SL, Pendergrast DG, McGhee CNJ. Changing Trends in Corneal Transplantation in Aotearoa/New Zealand, 1991 to 2020: Effects of Population Growth, Cataract Surgery, Endothelial Keratoplasty, and Corneal Cross-Linking for Keratoconus. Cornea. 2022 Jun 1;41(6):680-687. doi: 10.1097/ICO.0000000000002812. Epub 2021 Jul 14. PMID: 34267061.

Moriyama AS, Dos Santos Forseto A, Pereira NC, Ribeiro AC, de Almeida MC, Figueras-Roca M, Casaroli-Marano RP, Mehta JS, Hofling-Lima AL. Trends in Corneal Transplantation in a Tertiary Hospital in Brazil. Cornea. 2022 Jul 1;41(7):857-866. doi: 10.1097/ICO.0000000000002801. Epub 2021 Jul 21. PMID: 34294631.

In addition, although there is still not enough published evidence, several cases of failure have been reported in the literature, and the final endothelial densities seem to be very low, the alternative of Descemetorhexis Without Endothelial Keratoplasty should be mentioned and commented.

Additional references on Descemetorhexis Without Endothelial Keratoplasty:

Galvis V, Tello A, Berrospi RD, Cuadros MO, Blanco NA. Descemetorhexis Without Endothelial Graft in Fuchs Dystrophy. Cornea. 2016 Sep;35(9):e26-8. doi: 10.1097/ICO.0000000000000931. PMID: 27429088.

Iovieno A, Neri A, Soldani AM, Adani C, Fontana L. Descemetorhexis Without Graft Placement for the Treatment of Fuchs Endothelial Dystrophy: Preliminary Results and Review of the Literature. Cornea. 2017 Jun;36(6):637-641. doi: 10.1097/ICO.0000000000001202. PMID: 28410355.

Franceschino A, Dutheil F, Pereira B, Watson SL, Chiambaretta F, Navel V. Descemetorhexis Without Endothelial Keratoplasty in Fuchs Endothelial Corneal Dystrophy: A Systematic Review and Meta-Analysis. Cornea. 2022 Jul 1;41(7):815-825. doi: 10.1097/ICO.0000000000002855. Epub 2021 Dec 7. PMID: 34879044.

Furthermore, the two most recent publication by the Kinoshita´s group, should be also mentioned and discussed:

Numa K, Imai K, Ueno M, Kitazawa K, Tanaka H, Bush JD, Teramukai S, Okumura N, Koizumi N, Hamuro J, Sotozono C, Kinoshita S. Five-Year Follow-up of First 11 Patients Undergoing Injection of Cultured Corneal Endothelial Cells for Corneal Endothelial Failure. Ophthalmology. 2021 Apr;128(4):504-514. doi: 10.1016/j.ophtha.2020.09.002. Epub 2020 Sep 6. PMID: 32898516.

Ueno M, Toda M, Numa K, Tanaka H, Imai K, Bush J, Teramukai S, Okumura N, Koizumi N, Yamamoto A, Tanaka M, Sotozono C, Hamuro J, Kinoshita S. Superiority of Mature Differentiated Cultured Human Corneal Endothelial Cell Injection Therapy for Corneal Endothelial Failure. Am J Ophthalmol. 2022 May;237:267-277. doi: 10.1016/j.ajo.2021.11.012. Epub 2021 Nov 14. PMID: 34788595.

Finally, other previously published studies on magnetic attraction used to hold in position endothelial cells, should be also discussed.

Mimura T, Yamagami S, Usui T, Ishii Y, Ono K, Yokoo S, Funatsu H, Araie M, Amano S. Long-term outcome of iron-endocytosing cultured corneal endothelial cell transplantation with magnetic attraction. Exp Eye Res. 2005 Feb;80(2):149-57. doi: 10.1016/j.exer.2004.08.021. PMID: 15670793.

Cornell LE, Wehmeyer JL, Johnson AJ, Desilva MN, Zamora DO. Magnetic Nanoparticles as a Potential Vehicle for Corneal Endothelium Repair. Mil Med. 2016 May;181(5 Suppl):232-9. doi: 10.7205/MILMED-D-15-00151. PMID: 27168578.

Moysidis SN, Alvarez-Delfin K, Peschansky VJ, Salero E, Weisman AD, Bartakova A, Raffa GA, Merkhofer RM Jr, Kador KE, Kunzevitzky NJ, Goldberg JL. Magnetic field-guided cell delivery with nanoparticle-loaded human corneal endothelial cells. Nanomedicine. 2015 Apr;11(3):499-509. doi: 10.1016/j.nano.2014.12.002. Epub 2015 Jan 14. PMID: 25596075; PMCID: PMC4691344.

Line 181, it reads: “Twenty-five male New Zealand white rabbits 6-8 months old, weighing between 1.8 to 2.2 kg, were assigned randomly to two wound model groups and four experimental treatment groups.”

COMMENT

I am trying to understand how the distribution of the 25 animals in the experiments was made, but it is not easy, since the description of the allocation process is actually incomplete.

As I understand it, after rereading several times, two separate experiments were carried out sequentially, the first was the "Endothelial Wound Model" including 6 animals distributed in two groups (SW and DM) and the second experiment was the Experimental treatment with cultured cell injections. However, the numbers do not add up, since if 6 animals were used in the first experiment, 19 would remain for the second experiment, and only 4 groups each with 4 animals are mentioned.

Consider modifying to: “22 male New Zealand White rabbits 6 to 8 months old, weighing between 1.8 and 2.2 kg, were used for two different experiments, performed sequentially: in the first experiment 6 rabbits were randomly assigned to two endothelial wound model groups and then, in the second experiment, another 16 animals were assigned to four treatment groups”.

Line 194, it reads: “Three rabbits (n=3) were submitted to the classical "descemetorhexis" technique, as described by Melles et al[34].”

COMMENT

The adjective “classical” and the verb “submit” do not fit the sentence. If it is indicated that there are three animals, it is redundant to include the sample size in numbers in parentheses. The reference number (34) is wrong, it should read (33).

Therefore, the sentence should read. “Three rabbits were submitted to the "Descemetorhexis" technique, as originally described by Melles et al[33].”

Line. 199, it reads: “After the borders were released, the rest of the membrane was gently pulled centripetally towards the side-port incision with care to avoid trauma to the corneal stroma”.

COMMENT

This description of the technique is not really adequate.

Consider modifying to: “After the borders were released, the rest of the Descemet membrane was gently centripetally stripped off the posterior stroma and then pulled towards the side-port incision and removed from the anterior chamber, very carefully, to avoid trauma to the corneal stroma”.

Line 202, it reads: “For both groups, the corneal incisions were closed with 10-0 nylon sutures (Ethicon, Johnson & Johnson, Brazil) followed by an injection of 0.3 mL triamcinolone acetonide injection (4 mg/0,1mL, Ophtaac®, Ophthalmos, Brazil) and ciprofloxacin/dexamethasone drops (EMS, LeGrand, Brazil).”

COMMENT

The word "injection" is repeated. Was the injection of 0.3 mL triamcinolone acetonide intracameral? Furthermore, it should be clarified that ciprofloxacin/dexamethasone were used topically in the immediate postoperative period.

Line 227. It reads: “An adapted bandage contact lens was glued with”

COMMENT

It seems that this sentence is incomplete. What was used to glue the contact lens?

Line 228. It reads: “Antibiotic prophylaxis consisting of a single drop of 0.3% gatifloxacin/1% prednisolone acetate, immediately instilled after the procedure and maintained q.i.d postoperatively until euthanasia, at one month. The contralateral eye served as control. Clinical edema and inflammation were assessed daily for four days with a portable slit-lamp. After the euthanasia, eye globes from each group were enucleated and assessed with autofluorescence to check for cultured HCEC adhesion.”.

COMMENT

The sentences are not really clear. 

Consider modifying to: “A fixed combination of 0.3 % Gatifloxacin and 1% prednisolone acetate was applied topically immediately after the procedure and then q.i.d postoperatively until euthanasia was performed, one month after the intracameral injection. The contralateral eye served as a control. Clinical edema and inflammation were assessed daily for only four days with a portable slit lamp, but to assess cultured HCEC adhesion, eye globes from each group were enucleated and assessed with autofluorescence, after the euthanasia.”

Line 234. It reads: “The isolation and adhesion of in vitro cultured HCEC, ex vivo attachment of SPM embedded and non-SPM cultured HCEC, as well as in vivo experiments are summarized in Fig. 1”.

COMMENT

All of the components in Figure 1 are too small, making them really impossible to read. The sample size of each group 

The Figure should be greatly increased in size, so that all the components are easily understood, and the internal legends are legible. In addition, the first “in vivo” experiment (i.e., endothelia wound model, including two groups of three animals each) should be also included in the Figure. Furthermore, in section A of figure, it would be much clearer for the reader if instead of drawing a series of cell culture dishes, only one was drawn and the sample size (n= XX) was indicated. Sample sizes should also be indicated in the diagrams of all other experiments.

Line 285. It reads: “Shortly after the endothelial wound creation, intense corneal edema was a common feature in SW and DMRx groups at the 7th postoperative assessment”.

COMMENT

There seems to be a typo here. It is possible that the authors mean that in the first in vivo experiment with rabbits, that of the endothelial wound model, on the seventh day corneal edema was found in both groups. 

Consider modifying to: “Shortly after the endothelial wound creation, intense corneal edema appeared, and it was a common feature in SW and DMRx groups at the 7th postoperative day assessment”.

Line 290, it reads: “Moreover, the animals from the SW group appeared to have fully recovered endothelial function with completely transparent corneas and no signs of scarring at the final slit-lamp evaluation”.

COMMENT

It is important to indicate the time point of the final slit-lamp evaluation.

Therefore, it should read: “Moreover, the animals from the SW group appeared to have fully recovered endothelial function with completely transparent corneas and no signs of scarring at the final slit-lamp evaluation, 30 days postoperatively”.

In Figure 5 legend it is very important to state at what follow-up time were the evaluations performed, and also the number of eyes in each group.

Line 417, it reads: “Peh and Okumura preconize that either 2.000 cells/mm2[28] or at least 300.000 cells[30].should be injected into the anterior chamber to account for cell loss secondary to reflux, poor adhesion, or apoptosis.”

COMMENT

The sentence is confusing. Firstly, the number of cells injected should not be indicated in concentration, but in absolute numbers. Secondly, the range that is mentioned is too extreme, so there must be an error (it is indicated that at least 2000 cells/mm2 and then 300,000 cells/m2). Checking the study by Peh et al, they injected 600 000 cells into the anterior chamber.  In addition, the combination of "either", and "or at least", is not really suitable in the context. Finally, the reference 30 is not related to Peh or Okumura. Since this is not the only error in the cited references, all citations should be carefully checked to ensure that they agree with the reference that you actually want to cite, throughout the entire manuscript.

This sentence must be rewritten, confirming the amounts of endothelial cells actually injected in the studies to be mentioned, and the authors must be cited correctly, that is: Peh et al, Okumura et al, etc. If it is written as Peh and Okumura, it is as if they were the only two authors of a study, which is not correct. 

Line 445, it reads: “Finally, after electing the scrape wound as the appropriate model for challenging the functional aspect of our HCEC culture protocol, we compared edema resolution with cultured HCEC alone or embedded with SPM.”

COMMENT

I am trying to understand the structure of the different experiments, which has not been easy at all. At this point my concern is whether after selecting the scrape wound model as the appropriate one for the endothelial cell injection animal models, the cells that were injected, whether cultured HCEC alone or embedded with SPM, had been treated or not with Rho-kinase inhibitor (ROCK) inhibitor (Y27632)? This detail, which is critical, must be explicitly indicated in this section of the manuscript.

Line 447. It reads: “The short follow-up time (4 days) was explicitly selected because our observations with scraped wound experiments revealed maximum clinical edema at this time point. Therefore, we presumed that regeneration of rabbit endothelium was still at bay, and outcome analysis at this point would avoid bias.”.

COMMENT 

The words “the short follow-up time (4 days)” suggested that there were several options to choose in the protocol, which is not the case.  The adverb explicitly, which means: “in an explicit manner: clearly and without any vagueness or ambiguity”, does not fit the context. The idiom “at bay”, which is usually used with the verbs “keep” or “hold”, means that something or someone is in the position of being unable to move closer while attacking or trying to approach someone. This is idiom does not fit the context. It is necessary to include the citation of the previous observations in experiments with rabbits mentioned by the authors, or if they were not published, include the annotation: (unpublished results).

Therefore, consider modifying to: “We deliberately decided to employ a very short follow-up time (only 4 days), due to our previous observations from experiments in rabbit models with endothelial injury, in which we found that maximal edema was found at this time point (unpublished results). In this way we avoided the influence of the natural regeneration capacity of the rabbit endothelium, which at this early stage after trauma seems not yet to have started”.

Author Response

Please check this link for the answers in a word file: https://www.dropbox.com/s/zrg3hdekzg8gsqi/REVIEWER%20ROUND%201-%20Rev%202%20Queries%20FInal.docx?dl=0

REVIEWER 2

Comments and Suggestions for Authors

Initial Framework for Human Corneal Endothelial Cell Culture and Preliminary Wound Model Experiments.

INTRODUCTION

Q1 Line 32. It reads: “The human corneal endothelial cells (HCECs) are responsible for keeping the cornea transparent by maintaining the tissue at a semi-dry state (~70% hydration) through the action of their ionic pumps and tight junction barrier[1].. HCEC have negligible in vivo proliferation rate[2]., likely due to the aqueous humor high concentrations of TGF-2 and cAMP that up-regulate p27KIP1, thus preventing HCEC from proliferating then regenerate upon wounding[3, 4].”

COMMENT

It is true than under normal circumstances HCEC have negligible in vivo proliferation rate, but in special conditions it has been shown that the replication can increase.

Three of the cited references (2-4) were published 19 or more years ago, therefore, for this reason, they can actually be considered outdated.

Additional suggested references:

Galvis V, Tello A, Gutierrez ÁJ. Human corneal endothelium regeneration: effect of ROCK inhibitor. Invest Ophthalmol Vis Sci. 2013 Jul 24;54(7):4971-3. doi: 10.1167/iovs.13-12388. PMID: 23883789.

Zavala J, López Jaime GR, Rodríguez Barrientos CA, Valdez-Garcia J. Corneal endothelium: developmental strategies for regeneration. Eye (Lond). 2013 May;27(5):579-88. doi: 10.1038/eye.2013.15. Epub 2013 Mar 8. PMID: 23470788; PMCID: PMC3650267.

Klyce SD. Endothelial pump and barrier function. Exp Eye Res. 2020 Sep;198:108068. doi: 10.1016/j.exer.2020.108068. Epub 2020 Jul 11. PMID: 32663497.

Van den Bogerd B, Dhubhghaill SN, Koppen C, Tassignon MJ, Zakaria N. A review of the evidence for in vivo corneal endothelial regeneration. Surv Ophthalmol. 2018 Mar-Apr;63(2):149-165. doi: 10.1016/j.survophthal.2017.07.004. Epub 2017 Aug 4. PMID: 28782549.

Reply 1: Dear Reviewer, thank you for the comment. We have replaced the outdated references for two of the suggested updated references:

Klyce SD. Endothelial pump and barrier function. Exp Eye Res. 2020 Sep;198:108068. doi: 10.1016/j.exer.2020.108068. Epub 2020 Jul 11. PMID: 32663497.

Van den Bogerd B, Dhubhghaill SN, Koppen C, Tassignon MJ, Zakaria N. A review of the evidence for in vivo corneal endothelial regeneration. Surv Ophthalmol. 2018 Mar-Apr;63(2):149-165. doi: 10.1016/j.survophthal.2017.07.004. Epub 2017 Aug 4. PMID: 28782549.

Galvis V, Tello A, Gutierrez ÁJ. Human corneal endothelium regeneration: effect of ROCK inhibitor. Invest Ophthalmol Vis Sci. 2013 Jul 24;54(7):4971-3. doi: 10.1167/iovs.13-12388. PMID: 23883789.

Q2 Line 38. It reads: “When HCEC density falls below 500-700 cells/mm, the cornea is at a high risk of decompensating and eventually leads to blindness due to corneal fibrosis secondary to bullous keratopathy[1, 7].”

COMMENT

It would be very useful for the reader to point out here that in various areas of the world, endothelial failure is among the first indications for corneal transplantation.

Consider modifying to: “When HCEC density falls below 500-700 cells/mm, the cornea is at a high risk of decompensating requiring an endothelial corneal transplantation, but if the edema is chronic enough eventually may lead to stromal fibrosis secondary to bullous keratopathy and will require a penetrating corneal transplantation. [1, 7]. Currently bullous keratopathy due to endothelial failure is one of the first indications for corneal transplantation in many parts of the world.”

References to be cited:

Galvis V, Tello A, Laiton AN, Salcedo SLL. Indications and techniques of corneal transplantation in a referral center in Colombia, South America (2012-2016). Int Ophthalmol. 2019 Aug;39(8):1723-1733. doi: 10.1007/s10792-018-0994-z. Epub 2018 Jul 25. PMID: 30047076.

Zhao S, Wan X, Yao W, Xu J, Le Q. Trends of Corneal Transplantation in Adults from 2010 to 2019 in East China: A 10-Year Experience. Ophthalmic Res. 2022;65(1):30-39. doi: 10.1159/000519824. Epub 2021 Sep 29. PMID: 34587612.

Chilibeck CM, Brookes NH, Gokul A, Kim BZ, Twohill HC, Moffatt SL, Pendergrast DG, McGhee CNJ. Changing Trends in Corneal Transplantation in Aotearoa/New Zealand, 1991 to 2020: Effects of Population Growth, Cataract Surgery, Endothelial Keratoplasty, and Corneal Cross-Linking for Keratoconus. Cornea. 2022 Jun 1;41(6):680-687. doi: 10.1097/ICO.0000000000002812. Epub 2021 Jul 14. PMID: 34267061.

Moriyama AS, Dos Santos Forseto A, Pereira NC, Ribeiro AC, de Almeida MC, Figueras-Roca M, Casaroli-Marano RP, Mehta JS, Hofling-Lima AL. Trends in Corneal Transplantation in a Tertiary Hospital in Brazil. Cornea. 2022 Jul 1;41(7):857-866. doi: 10.1097/ICO.0000000000002801. Epub 2021 Jul 21. PMID: 34294631.

Reply 2: Thank you for the comment. We have included suggested references and changed this sentence to: “Currently bullous keratopathy due to endothelial failure is a leading cause of corneal transplantation in many parts of the world accounting for more than 40% of keratoplasty surgeries in some continents[8-10].” (Lines 43-46)

Moriyama AS, Dos Santos Forseto A, Pereira NC, Ribeiro AC, de Almeida MC, Figueras-Roca M, Casaroli-Marano RP, Mehta JS, Hofling-Lima AL. Trends in Corneal Transplantation in a Tertiary Hospital in Brazil. Cornea. 2022 Jul 1;41(7):857-866. doi: 10.1097/ICO.0000000000002801. Epub 2021 Jul 21. PMID: 34294631.

Galvis V, Tello A, Laiton AN, Salcedo SLL. Indications and techniques of corneal transplantation in a referral center in Colombia, South America (2012-2016). Int Ophthalmol. 2019 Aug;39(8):1723-1733. doi: 10.1007/s10792-018-0994-z. Epub 2018 Jul 25. PMID: 30047076. Currently bullous keratopathy due to endothelial failure is a leading cause for corneal transplantation in many parts of the world accounting for more than 40% of keratoplasty surgeries in some continents[8-10].

Q3 In addition, although there is still not enough published evidence, several cases of failure have been reported in the literature, and the final endothelial densities seem to be very low, the alternative of Descemetorhexis Without Endothelial Keratoplasty should be mentioned and commented.

Additional references on Descemetorhexis Without Endothelial Keratoplasty:

Galvis V, Tello A, Berrospi RD, Cuadros MO, Blanco NA. Descemetorhexis Without Endothelial Graft in Fuchs Dystrophy. Cornea. 2016 Sep;35(9):e26-8. doi: 10.1097/ICO.0000000000000931. PMID: 27429088.

Iovieno A, Neri A, Soldani AM, Adani C, Fontana L. Descemetorhexis Without Graft Placement for the Treatment of Fuchs Endothelial Dystrophy: Preliminary Results and Review of the Literature. Cornea. 2017 Jun;36(6):637-641. doi: 10.1097/ICO.0000000000001202. PMID: 28410355.

Franceschino A, Dutheil F, Pereira B, Watson SL, Chiambaretta F, Navel V. Descemetorhexis Without Endothelial Keratoplasty in Fuchs Endothelial Corneal Dystrophy: A Systematic Review and Meta-Analysis. Cornea. 2022 Jul 1;41(7):815-825. doi: 10.1097/ICO.0000000000002855. Epub 2021 Dec 7. PMID: 34879044.

Q4 Furthermore, the two most recent publication by the Kinoshita´s group, should be also mentioned and discussed:

Numa K, Imai K, Ueno M, Kitazawa K, Tanaka H, Bush JD, Teramukai S, Okumura N, Koizumi N, Hamuro J, Sotozono C, Kinoshita S. Five-Year Follow-up of First 11 Patients Undergoing Injection of Cultured Corneal Endothelial Cells for Corneal Endothelial Failure. Ophthalmology. 2021 Apr;128(4):504-514. doi: 10.1016/j.ophtha.2020.09.002. Epub 2020 Sep 6. PMID: 32898516.

Ueno M, Toda M, Numa K, Tanaka H, Imai K, Bush J, Teramukai S, Okumura N, Koizumi N, Yamamoto A, Tanaka M, Sotozono C, Hamuro J, Kinoshita S. Superiority of Mature Differentiated Cultured Human Corneal Endothelial Cell Injection Therapy for Corneal Endothelial Failure. Am J Ophthalmol. 2022 May;237:267-277. doi: 10.1016/j.ajo.2021.11.012. Epub 2021 Nov 14. PMID: 34788595.

Q5 Finally, other previously published studies on magnetic attraction used to hold in position endothelial cells, should be also discussed.

Mimura T, Yamagami S, Usui T, Ishii Y, Ono K, Yokoo S, Funatsu H, Araie M, Amano S. Long-term outcome of iron-endocytosing cultured corneal endothelial cell transplantation with magnetic attraction. Exp Eye Res. 2005 Feb;80(2):149-57. doi: 10.1016/j.exer.2004.08.021. PMID: 15670793.

Cornell LE, Wehmeyer JL, Johnson AJ, Desilva MN, Zamora DO. Magnetic Nanoparticles as a Potential Vehicle for Corneal Endothelium Repair. Mil Med. 2016 May;181(5 Suppl):232-9. doi: 10.7205/MILMED-D-15-00151. PMID: 27168578.

Moysidis SN, Alvarez-Delfin K, Peschansky VJ, Salero E, Weisman AD, Bartakova A, Raffa GA, Merkhofer RM Jr, Kador KE, Kunzevitzky NJ, Goldberg JL. Magnetic field-guided cell delivery with nanoparticle-loaded human corneal endothelial cells. Nanomedicine. 2015 Apr;11(3):499-509. doi: 10.1016/j.nano.2014.12.002. Epub 2015 Jan 14. PMID: 25596075; PMCID: PMC4691344.

Reply 3,4,5: Thank you for your comment. To avoid a too extensive introduction we have refrained from commenting all therapeutical options for CED in detail, but we agree mentioning these alternatives (DSO/DWEK, cell injection and strategies for cell positioning) might add clarity to the purpose of our study. We have modified this paragraph and drafted an additional one addressing these topics with the suggested references.

“The introduction of ROCKi as a supplement to HCEC culture has also aroused researchers to investigate it’s in vivo regenerative properties, which have proven beneficial in sur-geries that selectively remove the diseased endothelium (DSO/DWEK) [26,27]. The ul-timate outcome of selective endothelial removal techniques is still controversial, with several cases resulting in very low endothelial cell count or failure [28]. Meanwhile, two cell delivery methods have shown satisfactory and safe outcomes in several studies in-volving animal models that have progressed to clinical trials, namely cell injection [29–31] and tissue-engineered endothelial keratoplasty [32,33].” (Lines 73-80)

“Thus far, the cell injection therapy technique proposed by Kinoshita et al. relies solely on gravitational forces to promote cultured HCEC attachment to the recipient Descemet membrane (DM), which requires patients to withstand a long period in an uncomfortable prone position [34]. This method depends on patient compliance and is inherently subject to the inadvertent delivery of cells to areas that are completely outside of the targeted treatment (unwounded DM, iris, trabeculum, and lens) [35]; hence, a large number of cells (500,000/200 µL) need to be injected into the anterior chamber [36]. Iron microsphere or the nanoparticle embeddement of CEC with magnetic delivery has been described as an alternative method to increase in vitro delivery efficiency 2.4-fold compared to gravity [37] without changing their canonical traits [38] while reducing corneal edema in animal models [39].” (Lines 81-91)

A brief comment regarding the article by Ueno and Kinoshita et al. was also included at the introduction.

“The first clinical trial with cell injection supplemented with ROCKi in Japan unveiled satisfactory and fast patient recovery [34], with a survival rate of 90.9% after 5 years [40]. However, regulatory constraints in most countries impose the strict use of corneas that are unsuitable for clinical purposes or the peripheral remnants of from keratoplasty, which may not yield enough HCECs of good quality to restore corneal transparency [41].” (Lines 92-96)

METHODS

Q6 Line 181, it reads: “Twenty-five male New Zealand white rabbits 6-8 months old, weighing between 1.8 to 2.2 kg, were assigned randomly to two wound model groups and four experimental treatment groups.”

COMMENT

I am trying to understand how the distribution of the 25 animals in the experiments was made, but it is not easy, since the description of the allocation process is actually incomplete.

As I understand it, after rereading several times, two separate experiments were carried out sequentially, the first was the "Endothelial Wound Model" including 6 animals distributed in two groups (SW and DM) and the second experiment was the Experimental treatment with cultured cell injections. However, the numbers do not add up, since if 6 animals were used in the first experiment, 19 would remain for the second experiment, and only 4 groups each with 4 animals are mentioned.

Consider modifying to: “22 male New Zealand White rabbits 6 to 8 months old, weighing between 1.8 and 2.2 kg, were used for two different experiments, performed sequentially: in the first experiment 6 rabbits were randomly assigned to two endothelial wound model groups and then, in the second experiment, another 16 animals were assigned to four treatment groups”.

Reply 6: Thank you for your comment. This sentence was modified according to the reviewer suggestion. (Lines 220-224)

Q7 Line 194, it reads: “Three rabbits (n=3) were submitted to the classical "descemetorhexis" technique, as described by Melles et al[34].”

COMMENT

The adjective “classical” and the verb “submit” do not fit the sentence. If it is indicated that there are three animals, it is redundant to include the sample size in numbers in parentheses. The reference number (34) is wrong, it should read (33).

Therefore, the sentence should read. “Three rabbits were submitted to the "Descemetorhexis" technique, as originally described by Melles et al[33].”

Reply 7: This sentence was modified according to the reviewer comments to “The descemetorhexis technique, as previously described by Melles et al. [33], was performed in three rabbits.” (Lines 235-236)

Q8 Line. 199, it reads: “After the borders were released, the rest of the membrane was gently pulled centripetally towards the side-port incision with care to avoid trauma to the corneal stroma”.

COMMENT

This description of the technique is not really adequate.

Consider modifying to: “After the borders were released, the rest of the Descemet membrane was gently centripetally stripped off the posterior stroma and then pulled towards the side-port incision and removed from the anterior chamber, very carefully, to avoid trauma to the corneal stroma”.

Reply 8: This sentence was modified according to the reviewer suggestion. (Lines 240-243)

Q9: Line 202, it reads: “For both groups, the corneal incisions were closed with 10-0 nylon sutures (Ethicon, Johnson & Johnson, Brazil) followed by an injection of 0.3 mL triamcinolone acetonide injection (4 mg/0,1mL, Ophtaac®, Ophthalmos, Brazil) and ciprofloxacin/dexamethasone drops (EMS, LeGrand, Brazil).”

COMMENT

The word "injection" is repeated. Was the injection of 0.3 mL triamcinolone acetonide intracameral? Furthermore, it should be clarified that ciprofloxacin/dexamethasone were used topically in the immediate postoperative period.

Reply 9: The injection was performed at the subconjunctival space, to reduce the risk of inadvertent cultured HCEC outflow or displacement. The drops were started immediately after surgery.

This sentence was changed as follows:

“For both groups, the corneal incisions were closed with 10-0 nylon sutures (Ethicon, Johnson & Johnson, Brazil) followed by a subconjunctival injection of 0.3 mL triamcinolone acetonide (4 mg/0,1mL, Ophtaac®, Ophthalmos, Brazil), ciprofloxacin/dexamethasone drops (EMS, LeGrand, Brazil) were started immediately after the procedure.” (Lines 244-247)

Q10: Line 227. It reads: “An adapted bandage contact lens was glued with”

COMMENT

It seems that this sentence is incomplete. What was used to glue the contact lens?

Reply 10: Thank you for your comment. The glue we used was 100% ethyl-2-cyanoacrylate. This sentence was changed accordingly as follows:

“An adapted bandage contact lens was glued with 100% ethyl-2-cyanoacrylate (Loctite®, Henkel, Brasil)” (Lines 270-271)

Q11: Line 228. It reads: “Antibiotic prophylaxis consisting of a single drop of 0.3% gatifloxacin/1% prednisolone acetate, immediately instilled after the procedure and maintained q.i.d postoperatively until euthanasia, at one month. The contralateral eye served as control. Clinical edema and inflammation were assessed daily for four days with a portable slit-lamp. After the euthanasia, eye globes from each group were enucleated and assessed with autofluorescence to check for cultured HCEC adhesion.”.

COMMENT

The sentences are not really clear. 

Consider modifying to: “A fixed combination of 0.3 % Gatifloxacin and 1% prednisolone acetate was applied topically immediately after the procedure and then q.i.d postoperatively until euthanasia was performed, one month after the intracameral injection. The contralateral eye served as a control. Clinical edema and inflammation were assessed daily for only four days with a portable slit lamp, but to assess cultured HCEC adhesion, eye globes from each group were enucleated and assessed with autofluorescence, after the euthanasia.”

Reply 11: Thank you for your comment. This sentence was modified according to the reviewer suggestion. (Lines 271-277)

Q12: Line 234. It reads: “The isolation and adhesion of in vitro cultured HCEC, ex vivo attachment of SPM embedded and non-SPM cultured HCEC, as well as in vivo experiments are summarized in Fig. 1”.

COMMENT

All of the components in Figure 1 are too small, making them really impossible to read. The sample size of each group 

The Figure should be greatly increased in size, so that all the components are easily understood, and the internal legends are legible. In addition, the first “in vivo” experiment (i.e., endothelia wound model, including two groups of three animals each) should be also included in the Figure.

Furthermore, in section A of figure, it would be much clearer for the reader if instead of drawing a series of cell culture dishes, only one was drawn and the sample size (n= XX) was indicated. Sample sizes should also be indicated in the diagrams of all other experiments.

Reply 12: Thank you for your comment. A second reviewer asked inclusion of other drawings to clarify the methods, so we have separated this figure into 3, increasing their size and including the modifications as suggested. (Lines 280-294)

Q13: Line 285. It reads: “Shortly after the endothelial wound creation, intense corneal edema was a common feature in SW and DMRx groups at the 7th postoperative assessment”.

COMMENT

There seems to be a typo here. It is possible that the authors mean that in the first in vivo experiment with rabbits, that of the endothelial wound model, on the seventh day corneal edema was found in both groups. 

Consider modifying to: “Shortly after the endothelial wound creation, intense corneal edema appeared, and it was a common feature in SW and DMRx groups at the 7th postoperative day assessment”.

Reply 13: Thank you for your comment. The reviewer is correct, it is indeed a typo. This sentence was modified according to the reviewer suggestion. (Line 345-347)

Q14: Line 290, it reads: “Moreover, the animals from the SW group appeared to have fully recovered endothelial function with completely transparent corneas and no signs of scarring at the final slit-lamp evaluation”.

COMMENT

It is important to indicate the time point of the final slit-lamp evaluation.

Therefore, it should read: “Moreover, the animals from the SW group appeared to have fully recovered endothelial function with completely transparent corneas and no signs of scarring at the final slit-lamp evaluation, 30 days postoperatively”.

Reply 14: This sentence was modified according to the reviewer suggestion. (Line 353)

Q15: In Figure 5 legend it is very important to state at what follow-up time were the evaluations performed, and also the number of eyes in each group.

Reply 15: We have included the modifications as suggested. (Lines 406-416)

Q16: Line 417, it reads: “Peh and Okumura preconize that either 2.000 cells/mm2[28] or at least 300.000 cells[30].should be injected into the anterior chamber to account for cell loss secondary to reflux, poor adhesion, or apoptosis.”

COMMENT

The sentence is confusing. Firstly, the number of cells injected should not be indicated in concentration, but in absolute numbers. Secondly, the range that is mentioned is too extreme, so there must be an error (it is indicated that at least 2000 cells/mm2 and then 300,000 cells/m2). Checking the study by Peh et al, they injected 600 000 cells into the anterior chamber.  In addition, the combination of "either", and "or at least", is not really suitable in the context. Finally, the reference 30 is not related to Peh or Okumura. Since this is not the only error in the cited references, all citations should be carefully checked to ensure that they agree with the reference that you actually want to cite, throughout the entire manuscript.

This sentence must be rewritten, confirming the amounts of endothelial cells actually injected in the studies to be mentioned, and the authors must be cited correctly, that is: Peh et al, Okumura et al, etc. If it is written as Peh and Okumura, it is as if they were the only two authors of a study, which is not correct. 

Reply 16: Thank you for the comment, we have revised all references in the manuscript before and after the suggested changes. We use endnote as a reference manager, it is likely that some error might have occurred while transforming the references to plain text. The rationale behind the inclusion of 2.000 cells/mm2 (Peh’s reference - which refers to tissue engineered sheets with cultured HCEC) was to communicate a number that is more familiar to what ophthalmologists use in their clinical practice. However, we do agree that in the context of cell injection this sentence may be misleading and confusing, therefore we have removed Peh’s reference and changed the sentence as follows:

“Okumura et al. preconize that a minimum of 500.000 cells should be injected into the anterior chamber to account for cell loss secondary to reflux, poor adhesion, or apoptosis[36]”. (Lines 480-481)

Q17: Line 445, it reads: “Finally, after electing the scrape wound as the appropriate model for challenging the functional aspect of our HCEC culture protocol, we compared edema resolution with cultured HCEC alone or embedded with SPM.”

COMMENT

I am trying to understand the structure of the different experiments, which has not been easy at all. At this point my concern is whether after selecting the scrape wound model as the appropriate one for the endothelial cell injection animal models, the cells that were injected, whether cultured HCEC alone or embedded with SPM, had been treated or not with Rho-kinase inhibitor (ROCK) inhibitor (Y27632)? This detail, which is critical, must be explicitly indicated in this section of the manuscript.

Reply 17: For the experimental cell therapy with rabbits, all cultured HCEC (alone or SPM-embedded) were treated with Y27632 ROCK inhibitor. This sentence was changed to

“Lastly, after electing the scrape wound as the appropriate model for challenging the functional aspect of our HCEC culture protocol, we compared edema resolution with cultured HCEC supplemented with Y27632 delivered by gravity or magnetic attraction (SPM-embedded HCEC).” (Lines 508-511)

Q18: Line 447. It reads: “The short follow-up time (4 days) was explicitly selected because our observations with scraped wound experiments revealed maximum clinical edema at this time point. Therefore, we presumed that regeneration of rabbit endothelium was still at bay, and outcome analysis at this point would avoid bias.”.

COMMENT 

The words “the short follow-up time (4 days)” suggested that there were several options to choose in the protocol, which is not the case.  The adverb explicitly, which means: “in an explicit manner: clearly and without any vagueness or ambiguity”, does not fit the context. The idiom “at bay”, which is usually used with the verbs “keep” or “hold”, means that something or someone is in the position of being unable to move closer while attacking or trying to approach someone. This is idiom does not fit the context. It is necessary to include the citation of the previous observations in experiments with rabbits mentioned by the authors, or if they were not published, include the annotation: (unpublished results).

Therefore, consider modifying to: “We deliberately decided to employ a very short follow-up time (only 4 days), due to our previous observations from experiments in rabbit models with endothelial injury, in which we found that maximal edema was found at this time point (unpublished results). In this way we avoided the influence of the natural regeneration capacity of the rabbit endothelium, which at this early stage after trauma seems not yet to have started”.

Reply 18: Thank you for your comment. This follow-up time was determined by both our unpublished observations and mentioned by Okumura et al with their previous papers.

“We deliberately decided to employ a very short follow-up time (only 4 days), due to ours (unpublished) and other previous observations from experiments in rabbit models with endothelial injury, in which the maximal edema was found at this time point [31,36,66,68]. In this way we avoided the influence of the natural regeneration capacity of the rabbit endothelium, which at this early stage after trauma seems not to have started yet.” (Lines 511-516)

Reviewer 3 Report

In this manuscript, Bandeira and co-workers described two procedures for the isolation of human corneal endothelial cells (HCECs) suitable to achieve GMP grade cells for clinical use and therapy, “focusing on real-life conditions of a research lab in Latin America”. Research-grade corneas from donors were used for isolating the HCECs, and collagenase digestion proved to result in better preservation of cell morphology than EDTA treatment. Supplementation with the Rho-kinase inhibitor, Y-27632 resulted in an increase in the final cell yield. Wound healing experiments ex vivo (human corneas) and in vivo (endothelial regeneration in rabbit corneas) were performed, using Vybrant Grenn labeled HCECs. The quality of the cell cultures was assessed by light and electron microscopy, and OCT fluorescence was used to estimate the adhesion and growth of injected cells. Cultured HCECs effectively adhered to the human corneas ex vivo, and improved corneal oedema and opacity in the rabbit model of corneal endothelial dysfunction. Injection in vivo with HCEC alone or after previous embedding with superparamagnetic  iron oxide nanoparticles  was able to clear the corneas either partially or totally.

Based on their results, the authors concluded that  injection of HCECs obtained by either isolation procedure may be appropriate for treating corneal endothelial dysfunction and preventing corneal scaring.

No doubt, the authors did an intense experimental work and some of the results may potentially be of interest for scientists and clinicians in the field of ophthalmology. However, the manuscript is difficult to read, the rationale of the experimental design is often elusive, and several weak points are present, as reported below.

Major points

The Section Material and methods is often unclear or inconsistent as to the procedures followed, and crucial technical details are missing, thus making the repetition of the experiments problematic.

Lines 95-97: why the treatment with collagenase was performed in multiwall plates and the one with EDTA in Falcon tubes? Is there any technical reason?

Lines 115-116: “The least toxic and most effective digestion method was chosen for all subsequent 115 experiments”: the method that was actually used (collagenase A?) should be indicated.

Line 117: I guess that group A was for the cells cultured in the absence of Rho-kinase inhibitor, but this is not explained.

Line 119-120: “Seeding was performed .... Y27632”: sentence unclear to be rephrased

Lines 122-139: which is the rationale of using a pre-treatment with iron oxide nanoparticles?

Lines 134-135: the whole procedure of collection,  fixation, embedding, sectioning, staining and observation for transmission electron microscopy is missing.

Lines 136-139: It is not explained how were the trypsinized cells collected: by conventional centrifugation or by magnetic displacement?

Line 213: “...resting prone position for 30 min”, but in Figure 1, the rabbit is supine for 30 min.

Line 253: “..adhered to the uncoated acrylic well”  but at line 106 “glass 6-well plate” were used.

Line 263: owing to the very different morphology and behaviour onto the growing surface of the cells isolated with either collagenase or EDTA,  immunolabelling with specific markers would have been necessary to surely define these cells as HCECs and exclude contamination with corneal keratinocytes (as the authors admitted at line 474).

Line 275: the XTT assay “confirms the safety of SPM embedding in cultured HCEC”: even if a direct cytotoxic effect of nanoparticle internalization was excluded, based on the this assay, is there any evidence for the long-term biocompatibility of these nanoconstructs on HCECs? This would be vital for the use of such cells in the therapy. In addition, the XTT assay provides a generic evidence of cell viability but it cannot demonstrate that internalization of SPM nanoparticles did not affect the morphological features of HCECs (as erroneously reported at line 361); a careful analysis of the TEM micrographs (that are not described at all in the legend of Figure 3) would be mandatory to support the authors’ statement.

Line 334: why “infrared image” if the OCT imaging was performed with an excitation of 488 nm (line 175)?

Lines 369-370 and 376: was the donor age restricted to 50 years or below 60? Discussion and Materials and methods are not consistent.

Throughout the text, “autofluorescence” (which refers to the fluorescence emitted naturally by a biological substance: https://www.merriam-webster.com/dictionary/autofluorescence) is incorrectly used instead of fluorescence or fluorescence signal.

Minor points

Linen 16: HCEC should be spelled out the first time is used

Line 35: TGF-b2

Lines 39, 269, 418, 421, 423: mm2

Line 55: tryple should be TrypLE

Lines 93 and 106: 37°C

Lines 93, 106, 144 and 169: CO2

Line 104: spontaneously (not spontaneouslu)

Line 113: inhibitor is repeated twice

Line 129: 5x103

Line 156: human CECs should be HCECs

Line 166, 221, 363 and 460:  “Descemet” or “Descemet’s membrane” should always be “DM”

Line 194: “(Fig__)”: which Figure?

Lines 203-204: injection is repeated twice

Lines 360 and 383: assay not essay

Line 402: collagenase A2 or collagenase A1?

Line 409, 410 or 430: EnMT or EMT, as at line 63?

In conclusion, I believe that this paper is unsuitable for publication in the International Journal of Molecular Sciences.

In conclusion, I believe that this paper is unsuitable for publication in the International Journal of Molecular Sciences.

Author Response

REVIEWER 3

Thank you for taking the time in reviewing our paper with such effort and thoroughness. We appreciate each and every of your comments. We hope to have improved our paper based on your enquiries, in such a way that it is more suitable for IJMS publication. 

Comments and Suggestions for Authors

In this manuscript, Bandeira and co-workers described two procedures for the isolation of human corneal endothelial cells (HCECs) suitable to achieve GMP grade cells for clinical use and therapy, “focusing on real-life conditions of a research lab in Latin America”. Research-grade corneas from donors were used for isolating the HCECs, and collagenase digestion proved to result in better preservation of cell morphology than EDTA treatment. Supplementation with the Rho-kinase inhibitor, Y-27632 resulted in an increase in the final cell yield. Wound healing experiments ex vivo (human corneas) and in vivo (endothelial regeneration in rabbit corneas) were performed, using Vybrant Grenn labeled HCECs. The quality of the cell cultures was assessed by light and electron microscopy, and OCT fluorescence was used to estimate the adhesion and growth of injected cells. Cultured HCECs effectively adhered to the human corneas ex vivo, and improved corneal oedema and opacity in the rabbit model of corneal endothelial dysfunction. Injection in vivo with HCEC alone or after previous embedding with superparamagnetic iron oxide nanoparticles  was able to clear the corneas either partially or totally.

Based on their results, the authors concluded that injection of HCECs obtained by either isolation procedure may be appropriate for treating corneal endothelial dysfunction and preventing corneal scaring.

No doubt, the authors did an intense experimental work and some of the results may potentially be of interest for scientists and clinicians in the field of ophthalmology. However, the manuscript is difficult to read, the rationale of the experimental design is often elusive, and several weak points are present, as reported below.

Major points

The Section Material and methods is often unclear or inconsistent as to the procedures followed, and crucial technical details are missing, thus making the repetition of the experiments problematic.

Lines 95-97: why the treatment with collagenase was performed in multiwall plates and the one with EDTA in Falcon tubes? Is there any technical reason?

Reply: There is no methodological explanation for this choice. It was simply a means to facilitate logistic while seeding the cornea halves and to avoid confusion when seeding the isolated cells.

Lines 115-116: “The least toxic and most effective digestion method was chosen for all subsequent 115 experiments”: the method that was actually used (collagenase A?) should be indicated.

Reply: This sentence was changed to: “Collagenase A (1 mg/mL) digestion was the least toxic and most effective digestion approach, hence it was chosen for all subsequent experiments.” (Lines 141-143)

Line 117: I guess that group A was for the cells cultured in the absence of Rho-kinase inhibitor, but this is not explained.

Line 119-120: “Seeding was performed .... Y27632”: sentence unclear to be rephrased

Reply: We agree with the reviewer comment. Group A refers to cultured HCEC that were not treated with ROCK inhibitors.. We have changed this paragraph to clarify the methodology.

“For all the samples in group A, the isolation before HCEC culture followed the steps exactly as described above, while samples from group B were submitted to supplementation with 10 µM of Y27632 during the seeding step.” (Lines 145-147)

Lines 122-139: which is the rationale of using a pre-treatment with iron oxide nanoparticles?

Reply: One of the issues with cell therapy is that the current delivery method with cell injection requires a far greater number of cells than what is physiologically required for proper maintenance of the endothelial pump to keep the corneal stroma transparent. Another problem is that the gravitational technique requires that patients stay a minimum of 3 hours in prone position, which increases the operating time and patient discomfort. The rationale of using superparamagnetic iron particles is to improve the efficacy of cell adhesion and reduce the operating time, without inducing aggregation of the nanoparticles or cells. Maybe with proper standardization and improvement of this technique we will be able to treat a larger number of patients with the same number of cultured cells. The actual comparison between the efficacy of conventional cell injection and by magnetic delivery is subject of future research in our lab, which is pending funds to be carried out.

Lines 134-135: the whole procedure of collection, fixation, embedding, sectioning, staining and observation for transmission electron microscopy is missing.

Reply: The description for the TEM was added in detail as follows.

“The successful embedding of HCEC with the SPM solution was evaluated by transmission electron microscopy (TEM) to confirm intracellular iron content. In brief, for this analysis cultured HCEC were seeded in glass slides and after 24h of SPM embedding they were fixed with 4% paraformaldehyde/1% glutaraldehyde in 0.1M phosphate buffer, stained with 0.5% uranyl acetate (en bloc), and then dehydrated in graded alcohol concentrations. Samples were transferred to propylene oxide and were embedded in Araldite resin. Ultrathin sections were collected on uncoated G300 copper grids and stained with 1% aqueous phosphotungstic acid and uranyl acetate. Sections were examined with a transmission electron microscope that operates between 80 and 120 kV (JEOL 1200 EX II; JEOL, Tokyo, Japan) equipped with a digital camera (GATAN 832, Pleasanton, CA).” (Lines 167-176)

Lines 136-139: It is not explained how were the trypsinized cells collected: by conventional centrifugation or by magnetic displacement?

Reply: the cells were collected by conventional centrifugation, as we do not have the proper device for MACS. The sentence was changed to elucidate this gap:

“First, the samples were incubated with the 0.1 Fe-mg/mL diluted SPM for 4 hours, the cultured media was removed, and each well was rinsed twice with PBS, followed by centrifugation as described in section 2.1.“ (Lines 160-162)

Line 213: “...resting prone position for 30 min”, but in Figure 1, the rabbit is supine for 30 min.

Reply: the figure was changed accordingly. (Line 295)

Line 253: “..adhered to the uncoated acrylic well”  but at line 106 “glass 6-well plate” were used.

Reply: in reviewing the article we have noted this was mistakenly described. All plates used in our research were acrylic. We have corrected this sentence. (Line 133)

Line 263: owing to the very different morphology and behaviour onto the growing surface of the cells isolated with either collagenase or EDTA, immunolabelling with specific markers would have been necessary to surely define these cells as HCECs and exclude contamination with corneal keratinocytes (as the authors admitted at line 474).

Reply: Although some type of contamination from other cell types (in this case, stromal keratocytes) is possible, our methodology previously associates the removal of the Descemet`s Membrane–Endothelial Cells monolayer, by using the descemetorrhexis technique. This measure considerably mitigates the possible contamination of other corneal cell types. Likewise, the culture medium used in this work is specific to facilitate the growth of corneal endothelial cells.

However, in parallel, and in collaboration with our associate Barcelona group, the collagenase isolation technique has been tested and the characterization and functionality of the cell cultures obtained have been assayed. The results obtained (not shown in this publication) are part of a future publication which does not involve SPM embedding nor animal work,

We carried out immunocytochemistry at isolation and at the first passage. Conventional primary antibodies used: mouse anti-ZO-1 antibody (5 µg/mL, Invitrogen, Thermo Fisher Scientific, Waltham, Massachusetts, USA), mouse anti-Na+/K+-ATPase α antibody (5 µg/mL, Merck; Darmstadt, Germany). At 21 days of cultures, positive cell monolayer was observed for both markers. ZO-1 positive demonstrated the formation of intercellular adhesions to maintain mosaic morphology. The presence of ATPase suggested certain functional activity. (These images are present at another manuscript that has been submitted to peer review and is currently under examination by the editorial board)

However, Proulx et al have recently published an article that specifically analyzed the digestion step of HCEC culture, in which they confirm our findings that collagenase is indeed a superior method for HCEC digestion compared to EDTA, with a more in-depth analyses that include immunostaining and functional analysis (TEER).

Santerre, K.; Proulx, S. Isolation efficiency of collagenase and EDTA for the culture of corneal endothelial cells. Mol Vis 2022, 28, 331-339.

Line 275: the XTT assay “confirms the safety of SPM embedding in cultured HCEC”: even if a direct cytotoxic effect of nanoparticle internalization was excluded, based on this assay, is there any evidence for the long-term biocompatibility of these nano-constructs on HCECs? This would be vital for the use of such cells in the therapy. In addition, the XTT assay provides a generic evidence of cell viability, but it cannot demonstrate that internalization of SPM nanoparticles did not affect the morphological features of HCECs (as erroneously reported at line 361); a careful analysis of the TEM micrographs (that are not described at all in the legend of Figure 3) would be mandatory to support the authors’ statement.

Reply: We agree with the reviewer that our statement may be misleading, hence we have rewritten this paragraph detailing the TEM results in separate, because the sole purpose of the TEM was to confirm that SPM particles were successfully embedded into the HCEC.

“The TEM confirmed successful SPM embedding with dark vacuoles located within cultured HCEC cytoplasms (Figure 5A). We observed that SPMs were endocytosed into HCECs without affecting cell-wall integrity or cytoplasmatic volume, and there was no leakage of SPM to the extracellular space.” (Lines 330-332)

As for the question if there is long-term biocompatibility of the HCEC nanoconstructs, it is out of the scope of this research paper at this point to evaluate the safety of this cell therapy approach. Nevertheless, 1 year after rabbit corneal endothelial cell transplantation facilitated by iron particles, Mimura et al. demonstrated the absence of ocular toxicity.

“No significant differences of the ERG (a- and b-wave amplitudes, and b-wave/a-wave ratio) were detected among the groups. Iron powder was not detected by Berlin blue staining in the ocular tissues of the RCEC-iron group. Apoptotic cells were not observed in the endothelium by terminal transferase-mediated nick-end labeling. Transplanted iron-endocytosing RCEC remained viable for 12 months after surgery. There were no detectable ocular complications after the transplantation of iron-endocytosing cultured RCEC.”

(Mimura, T.; Yamagami, S.; Usui, T.; Ishii, Y.; Ono, K.; Yokoo, S.; Funatsu, H.; Araie, M.; Amano, S. Long-term outcome of iron-endocytosing cultured corneal endothelial cell transplantation with magnetic attraction. Experimental eye research 2005, 80, 149-157, doi:10.1016/j.exer.2004.08.021.)

Meanwhile, Moysidis et al. noticed that the cell loading of SPM particles may be a temporary event

a time-dependence for endocytosis of the nanoparticles, with a twenty-four hour incubation period appearing to be optimal. Cells that were allowed to incubate with nanoparticles for one month had significantly fewer intracellular nanoparticles, suggesting that nanoparticles initially endocytosed are then removed from the cell during that time period. This may be due to a net exocytosis, or lysosomal degradation and removal of the nanoparticle contents from the cell, or some other mechanism.”  

(Moysidis, S.N.; Alvarez-Delfin, K.; Peschansky, V.J.; Salero, E.; Weisman, A.D.; Bartakova, A.; Raffa, G.A.; Merkhofer, R.M., Jr.; Kador, K.E.; Kunzevitzky, N.J.; et al. Magnetic field-guided cell delivery with nanoparticle-loaded human corneal endothelial cells. Nanomedicine 2015, 11, 499-509, doi:10.1016/j.nano.2014.12.002.)

In fact, since we used a much lower dose of SPM particles than what these authors described may actually increase the safety profile of the cellular approach proposed in the manuscript.

Line 334: why “infrared image” if the OCT imaging was performed with an excitation of 488 nm (line 175)?

Reply: According to Vybrant® DyeCycle® Green user guide (https://assets.fishersci.com/TFS-Assets/LSG/manuals/mp35004.pdf) “These stains take advantage of the commonly available 488 nm excitation source, placing cell cycle studies on live cells within reach of all flow cytometries. Vybrant® DyeCycleTM Green stain is excited at 488 nm with emission ~520 nm”

Lines 369-370 and 376: was the donor age restricted to 50 years or below 60? Discussion and Materials and methods are not consistent.

Reply: actually, the initial plan was to use corneas below 60, but we were able to procure corneas that were younger than 50 years old for all experiments. In order to clarify this confusion we have changed the sentence in methods section to “(…) under preservation was below ten days, and donor age below 50 years old.” (Line 109)

Throughout the text, “autofluorescence” (which refers to the fluorescence emitted naturally by a biological substance: https://www.merriam-webster.com/dictionary/autofluorescence) is incorrectly used instead of fluorescence or fluorescence signal.

Reply: the term autofluorescent throughout the paper refers to the “autofluorescent mode” of the OCT devices, which is originally intended to detect natural fluorescence of retinal structures. We have changed the sentences in which the term autofluorescence appears to clarify this mistake.

“The confocal scanning laser ophthalmoscopy (CSLO) used in the auto-fluoresce detection mode of the OCT was set at 488 nm and 521 nm wavelengths to detect Vybrant green for excitation and barrier phases, respectively.” (Lines 214-216)

“Conversely, for both NPT and positive control groups, the CLSO allowed the visualization of hyperreflective fluorescent points at the posterior layer of the wounded corneas (Fig.3, C), which confirmed the successful attachment of the cultured HCEC.” (Lines 338-341)

“The in vivo tracking with Vybrant Green was notably effective, as the fluorescence of labeled cultured HCEC could easily be discriminated by the CLSO OCT analysis.” (Lines 372-373)

“Lower row, CLSO OCT imaging system, inserts show the infra-red macroscopic image of the cornea” (Line 410)

“This accessible and safe cell-permeable DNA dye emits a 482 nm fluorescence that can easily be detected by the CLSO available in several imaging devices, commonly used in routine ophthalmology practice.” (Lines 501-504)

“Xia et al. has also described a similar method of cultured HCEC fluorescence tracking.”

Minor points

Line 16: HCEC should be spelled out the first time is used

Reply: we have rephrased this sentence to “Here, we describe a detailed framework from in vitro culture of human corneal endothelial cells (HCEC) to (…)” (Lines 16-17)

Line 35: TGF-b2

Reply: we have rephrased this sentence to “(…) likely due to the aqueous humor high concentrations of TGF-ß2 and cAMP (…)” (Line 37)

Lines 39, 269, 418, 421, 423: mm2

Reply: the connotation was rectified in all sentences

Line 55: tryple should be TrypLE

Reply: the connotation was rectified.

Lines 93 and 106: 37°C

Reply: the connotation was rectified.

Lines 93, 106, 144 and 169: CO2

Reply: the connotation was rectified.

Line 104: spontaneously (not spontaneouslu)

Reply: the sentence was rectified accordingly.

Line 113: inhibitor is repeated twice

Reply: the sentence was rectified to “The expansion of HCEC under ROCKi (Y27632; StemCell Technologies INC, Canada) was carried out after standardization of the HCEC isolation.” (Lines 140-141)

Line 129: 5x103

Reply: the connotation was rectified.

Line 156: human CECs should be HCECs

Reply: the abbreviation was rectified.

Line 166, 221, 363 and 460:  “Descemet” or “Descemet’s membrane” should always be “DM”

Reply: the abbreviation was rectified throughout the manuscript.

Line 194: “(Fig__)”: which Figure?

Reply: this was a reference to a figure that is only presented at the results section. Hence it has been removed from the methods part.

Lines 203-204: injection is repeated twice

Reply: this sentence was changed to “(…)followed by a subconjunctival injection of 0.3 mL triamcinolone acetonide(…)” (Line 246)

Lines 360 and 383: assay not essay

Reply: the word essay was substituted for assay in both sentences.

Line 402: collagenase A2 or collagenase A1?

Reply: The collagenase used was collagenase A (1 mg/mL), this was rectified along the manuscript.

Line 409, 410 or 430: EnMT or EMT, as at line 63?

Reply: we have corrected this typo and changed EMT to EnMT.

Round 2

Reviewer 1 Report

I mentioned in my earlier assessment that I did not find the present manuscript to be particularly novel. What is novel about the current manuscript if it has already been used clinically, as the authors themselves mentioned?
In addition, the authors rephrase the study's objectives independent of the site of the investigation.

The authors mentioned existing HCEC culture problems such as interdonor variability and endothelial mesenchymal transition.
The authors made no mention of them in their research methodologies or findings.

Author Response

Dear reviewer,

Question 1: I mentioned in my earlier assessment that I did not find the present manuscript to be particularly novel. What is novel about the current manuscript if it has already been used clinically, as the authors themselves mentioned?

Reply 1: As we mentioned in our previous reply (please see attachment thoroughly) there are significant differences between the methods employed by us and other pre-clinical studies. The methods described in our study have not been previously used clinically in humans neither by us nor by any other authors. Endothelial cell injection embedded with magnetic particles has been tested in animals in two studies (also detailed in our previous answers) but with completely different methods for culture, nanoparticle loading and wound models. In addition, the major novelty in  our manuscript is the method for non-invasive in vivo tracking (CLSO fluorescence) of cultured HCEC to confirm adhesion (as mentioned in the discussion of the earliest manuscript within the first submission).  

Question 2: In addition, the authors rephrase the study's objectives independent of the site of the investigation.

Reply: We apologize, but we cannot fully understand what is the exact query of the reviewer. Did the reviewer mean that we should rephrase/rewrite the study objectives removing the information that the study was conducted in Latin-america?

Question 3: The authors mentioned existing HCEC culture problems such as interdonor variability and endothelial mesenchymal transition.
The authors made no mention of them in their research methodologies or findings.

Reply 3: We did not include any methods for circumventing inter-donor variability in our methods or findings for several reasons.

First, procuring donor corneas to perform cell culture study is extremely challenging because there is the ethical paradox that "good quality" corneas must be directed to patients in demand for keratoplasty instead of clinical research.

Second, apart from a young donor age (whose exact range for proper culture is not exactly described in previous papers), there is a lack of knowledge regarding which other variables might influence the success/failures of cell culture protocols and what mechanism is related to these events.

Third, assuming that certain variables (e.g. cause of death, general health state, nutritional, previous eye surgeries) can impact the success of cell culture, if we tried to control these variables, this would increase costs immensely and for logistic reasons (expiration of consumables and reagents, animal maintenance) it would be impossible to gather enough samples to complete the study within an adequate time-frame and a reasonable budget.

Nevertheless, we did try to control significant variables (as mentioned in methods and discussion) by restricting donor-age below 50 years-old and ensuring a very fast delivery of the procured corneas via a direct line with the tissue bank (as mentioned in our discussion).

As for the mesenchymal transition, this event is typically troublesome after the passaging procedures and becomes more frequent with more attempts of exponential propagation (more than 2-3 passages). We did mention in both results and discussion that there were no signs of such phenomenon during cell expansion, which might be related to our variable control strategy and the fact that we used HCEC derived from primary cultures. Further phenotypical confirmation of cell quality and absence of this cell transition state are out of the scope of this study, but will be addressed in a subsequent study that is currently under review for publication.

Question 4:  (x) Moderate English changes required 
Reply: In the first round of reviewer queries, the reviewer pointed out extensive english and style changes. We did submit the paper to a paid review by a professional certified editor (as recommended by MDPI platform, please see the attached file). The certified english reviewer did not point out any problems with flow, clarity, conciseness, engament or delivery. His corrections were in regard to punctuations, some typos and referencing style.
We kindly ask that reviewer 1 points out which exactly are the issues he finds relevant for correction, so we can further improve this manuscript to make it suitable and worthy to IJMS readers. 

Thank you for your comments

Reviewer 2 Report

Very interesting study. All the mentioned issues were appropriately addressed. 

Author Response

Thank you for the comments and suggestions, we feel that after your inputs there was a significant improvement in our manuscript that will benefit IJMS subscribers.

Reviewer 3 Report

After the authors' revision, the overall paper quality was improved, thus making the manuscript suitable for publication.

Author Response

(The authors gave the same response as above.)

Round 3

Reviewer 1 Report

I understand the authors' attempts to explain that they employed alternative approaches; but, if the methods do not answer or are not examined current difficulties, such as inter-donor variability or EMT, what is the benefit of these new approaches?
If the authors just want to discuss state new methods, I recommend that they reframe the introduction and manuscript accordingly.

Author Response

Dear reviewer, thank you for your comments. The benefits of magnetic delivery with cell injection of cultured HCEC are discussed at the 6th paragraph of our introduction.

"Thus far, the cell injection therapy technique proposed by Kinoshita et al. relies solely on gravitational forces to promote cultured HCEC attachment to the recipient Descemet membrane (DM), which requires patients to withstand a long period in an uncomfortable prone position. This method depends on patient compliance and is inherently subject to the inadvertent delivery of cells to areas that are completely outside of the targeted treatment (unwounded DM, iris, trabeculum, and lens); hence, a large number of cells (500,000/200 µL) need to be injected into the anterior chamber. Iron microsphere or the nanoparticle embeddement of CEC with magnetic delivery has been described as an alternative method to increase in vitro delivery efficiency 2.4-fold compared to gravity without changing their canonical traits while reducing corneal edema in animal models."

In addition, we have made some changes to the manuscript according to your suggestions. They are listed below:

Title: A Framework for Human Corneal Endothelial Cell Culture and Preliminary Wound Model Experiments with a New Cell Tracking Approach. (Lines 1-4.)

Abstract: 

Here, we describe a detailed framework obtained from in vitro culture of human corneal endothelial cells (HCECs) and comparative in vivo experimental models for CED treatment with a new cell tracking approach. (Lines 17-19).

Ex vivo and in vivo adhesion with both cell delivery systems was confirmed by cell tracker fluorescence detection. (Lines 26-27)

Introduction: we have included sentences at the end of the sixth and seventh paragraph to highlight the cell delivery method approach and cell tracking novelty. Nevertheless, confirming a superior CEC adhesion in vivo with magnetic delivery methods after surgery remains challenging and is yet to be proven. (Lines 92-93)

We also investigated a wound model comparing HCEC attachment by magnetic attration with the prone position (gravitational) and a novel cell tracking method. (Lines 101-103)

Methods: we have changed names of  sub-sections 2.4 and 2.5.2 to emphasize the cell delivery approach and tracking method

2.4. Fluorescence Confirmation of Cultured HCEC Attachment to Recipient Corneas: Ex Vivo Wound Model (Lines 181-182)

2.5.2.  Experimental Treatment with Cultured Cell Injections and Fluorescence Confirmation of HCEC Adhesion (Lines 257-258)

Results: we have created a new subsection (3.1.5) to highlight the cell tracking approach

3.1.5. Fluorescence Tracking of Adherent Cultured HCEC after Injections

The in vivo tracking with Vybrant Green was notably effective, as the fluorescence of labeled cultured HCEC could easily be discriminated by the CSLO analysis. Distinctive fluorescence was detected centrally in the corneas of the NANO30 group (Figure 7D). In the HCEC120 group, the signal was heterogeneously distributed along the periphery and central area (Figure 7G), whereas it was completely absent in both control groups (Figure 7J,M).  (Lines 377-283)

Discussion: we have reframed the last paragraph (lines 548 to 557) at the end of the discussion, and included a couple of sentences explaining the potential benefits the cell tracking approach

In addition, we presented a simple and affordable method for cell tracking relying on a device that is already approved for clinical use (OCT). To our knowledge, this approach is the first of its kind and may be further improved to facilitate objective comparisons of cell adhesion efficacy between different delivery methods for CED (eg. counting of adherent cells). (Lines 550 to 554)

Conclusion: we have included a sentence before the closing statement regarding our findings with the cell tracking approach.

Furthermore, the discovery of a new cell tracking method may be of use to analyze and increase the efficacy of cell delivery systems for CED. (Lines 563 to 564)
